# Detecting hydrological connectivity using causal inference from time-series: synthetic and real karstic case studies

Damien Delforge[1,2], Olivier de Viron[3], Marnik Vanclooster[1], Michel Van Camp[2], and Arnaud Watlet[4]

[1]Earth and Life Institute, Université catholique de Louvain, Louvain-la-Neuve, Belgium
[2]Royal Observatory of Belgium, Brussels, Belgium
[3]Littoral, Environnement et Sociétés, Université de La Rochelle and CNRS (UMR7266), La Rochelle, France
[4]British Geological Survey, Nottingham, UK

**Correspondence:** Damien Delforge (damien.delforge@uclouvain.be)

**Abstract.** We investigate the potential of causal inference methods (CIMs) to reveal hydrological connections from time-series. Four CIMs are selected from two criteria, linear or nonlinear and bivariate or multivariate. A priori, multivariate and nonlinear CIMs are best suited for revealing hydrological connections because they fit nonlinear processes and deal with confounding factors such as rainfall, evapotranspiration, or seasonality. The four methods are applied to a synthetic case and a real karstic case study. The synthetic experiment confirms our expectation: unlike the other methods, the multivariate nonlinear framework has a low false-positive rate and allows for ruling out a connection between two disconnected reservoirs forced with similar effective precipitation. However, for the real case study, the multivariate nonlinear method was unstable because of the uneven distribution of missing values affecting the final sample size for the multivariate analyses, forcing us to cope with the results' robustness. Nevertheless, if we recommend a nonlinear multivariate framework to reveal actual hydrological connections, all CIMs bring valuable insights into the system's dynamics, making them a cost-effective and recommendable comparative tool for exploring data. Still, causal inference remains attached to subjective choices, operational constraints, and hypotheses challenging to test. As a result, the robustness of the conclusions that the CIMs can draw always deserves caution, especially with real, imperfect, and limited data. Therefore, alongside research perspectives, we encourage a flexible, informed, and limit-aware use of CIMs, without omitting any other approach that aims at the causal understanding of a system.

## 1 Introduction

Causal inference methods (CIMs) aim at identifying causal interactions between variables from data only (Spirtes et al., 2000; Pearl, 2009). When applied to time-series, these empirical methods are built upon the principle of priority of the cause, which goes back to Hume (Hume, 1748): CIMs infer causation from the expected time-dependencies, i.e., the causes preceding the effects. They have evolved throughout the 20[th] century to go beyond the well-known correlation, or cross-correlation, between two time-series (see Runge et al., 2019a, for a broad review). Although widely used, the correlation or cross-correlation method cannot identify nonlinear causal relationships nor discriminate actual causal links from dependencies resulting from confounding factors. Indeed, the common cause's principle (Reichenbach, 1956; Runge et al., 2019a) tells us that dependencies between two variables could result from a third confounding cause influencing both. Nowadays, many CIMs have been de-

veloped, differing in hypotheses or application fields. Some CIMs explicitly deal with, either or both, nonlinear dependencies or confounding factors through multivariate analyses. These new CIMs are of growing interest in Earth, land, and hydrological sciences (Meyfroidt, 2016; Runge et al., 2019a; Goodwell et al., 2020). In hydrology, applications of CIMs remain rare and cover, for example, the potential causal feedbacks between soil moisture and precipitation (Salvucci et al., 2002; Tuttle and Salvucci, 2017; Wang et al., 2018), the differential effects of environmental drivers of evapotranspiration (Ombadi et al., 2020), interactions across rainfall time scales (Molini et al., 2010), the ecohydrological feedback processes (Ruddell and Kumar, 2009), or the study of hydrological connectivity (Sendrowski and Passalacqua, 2017; Rinderer et al., 2018).

Our study compares four CIMs on hydrological case studies in the spirit of other recent comparative studies (Rinderer et al., 2018; Ombadi et al., 2020). Our application differs in the choice of CIMs and the research questions. We selected four CIMs, all operating on the time-domain following the principle of priority up to a maximum causal delay $d_{max}$. We use two bivariate CIMs, the linear Cross-Correlation Function (CCF), the nonlinear Convergent Cross Mapping (CCM) method (Sugihara et al., 2012; Ye et al., 2015), together with two multivariate methods, one linear testing for Partial Correlation (ParCorr) and one nonlinear testing for Conditional Mutual Information (CMI). The multivariate CIMs are parts of the same causal inference framework, PCMCI, a sequential procedure based on the PC algorithm (Spirtes and Glymour, 1991), followed by a test for Momentary Conditional Independence (MCI) (Runge et al., 2019b).

Like Rinderer et al. (2018), we discuss if CIMs are suitable for studying hydrological connectivity, which aims at identifying the paths taken or that could be taken by water (Bracken et al., 2013). We align with Rinderer et al. (2018)'s terminology inspired by the field of neurology and brain connectivity (Friston, 2011). Accordingly, there are three types of connectivity: (i) structural, (ii) functional, and (iii) effective connectivity. The structural connectivity is derived from the medium and highlights the potential, static, and time-invariant water flow paths from the geological environment's topography, spatial adjacency, or contiguity. The functional connectivity is dynamic and retrieved from statistical time-dependencies between local hydrological variables. Functional connectivity is a matter of cross-predictability and reflects dynamic links between the variables. These dynamic links are potential connections subject to confounding factors, i.e., they may or may not be related to a flow process between variables. Effective connectivity precisely refers to actual connections linked through hydrological processes and flows.

Rinderer et al. (2018) reviewed a broad ensemble of CIMs from the literature and tested them to assess groundwater connectivity. Like Ombadi et al. (2020), they report spurious effective connections (or False Positives), highlighting the imperfect, yet, variable detection performances of CIMs. To deal with False Positives, Rinderer et al. (2018) proposed to constrain or limit the results by an assessment of structural connectivity. However, our concern is the hydrological connectivity in the vadose zone of karst systems. Assessing structural connectivity in karst systems is a challenging task because of their hidden and heterogeneous structure (Bakalowicz, 2005). Thus, without structural connectivity assessment, we investigate hydrological connectivity from CIMs alone. Besides, karst systems are known for their nonlinear behavior, which could be imputed to nonlinear hydrological processes, e.g., taking the form of power laws, or threshold effects triggering flows (Bakalowicz, 2005; Blöschl and Zehe, 2005).

As a result, by design, multivariate nonlinear CIMs (e.g., PCMCI-CMI) would seem best suited to retrieve effective hydrological connections: they account for nonlinear dependencies and deal with confounding effects, e.g., seasonality or the forcing of precipitation and evapotranspiration, through a multivariate framework. Following a theoretical introduction of the different CIMs, we test this assertion and get hands-on the CIMs using a toy model to conduct a virtual experiment reproducing a simple case of two parallel hydrological reservoirs forced by the same effective precipitation. In a real case study, we apply the CIMs on data acquired at the Lorette cave (Rochefort, Southern Belgium). This dataset includes rainfall and potential evapotranspiration data, electrical resistivity time-series patterns from the subsurface obtained from a geophysical monitoring experiment using time-lapse Electrical Resistivity Tomography (ERT) (Watlet et al., 2018; Delforge et al., 2020b), and drip discharge time-series with distinct dynamical patterns monitoring percolation at three spots within the cave. In particular, previous dye tracing tests have revealed fast connected preferential flow between the surface and a particular spot in the cave (Poulain et al., 2018). We expect CIMs to reveal this specific connection. Time-series also have different numbers of missing values, unevenly distributed over time, allowing a discussion of the impact of temporal gaps or short sample size on the analysis.

## 2    Materials and Methods

### 2.1    Causal Inference Methods (CIMs)

#### 2.1.1    Cross-Correlation Function (CCF)

With CCF, we consider that a variable $X_t$ is said to be a cause of variable $Y_t$ if the Pearson's correlation coefficient $\rho$ between $Y_t$ and $X_{t-d}$ is significant on their overlapping domain for at least one value of $d$ up to $d_{max}$. The significance is assessed with a Student's t-test (see supplementary materials SM1 for more details on the CIMs' implementation).

Usually, the CCF method is not explicitly presented as a CIM. Nevertheless, we consider it as such because the method is simple, intelligible as linearly interpretable, and, in practice, widely used for the implicit purpose of making causal inference, despite its limits as a bivariate linear method, in most scientific domains like hydrology (e.g., Angelini, 1997; Larocque et al., 1998; Labat et al., 2000; Mathevet et al., 2004; Bailly-Comte et al., 2008; Watlet et al., 2018). In general, linear methods have been popular in hydrology and attractive for their computational efficiency (Dooge, 1973). Besides, as a result of CCF popularity, the outcomes of any other CIMs are best appreciated when benchmarked against the well-known CCF method.

#### 2.1.2    Convergent Cross-Mapping (CCM)

CCM is primarily designed to reveal weak nonlinear interactions between time-series (Sugihara et al., 2012; Ye et al., 2015). CCM is a nearest-neighbor forecasting approach. It tests whether dynamic trajectories between two variables behave consistently, i.e., with some cross-predictive skills, while the system revisits the same states (dynamic recurrence). The system states are usually unknown; they are approximated by trajectory segments found in a trajectory matrix $M_Y$ given by the Takens'

embedding theorem (Takens, 1981):

$$M_Y = \{Y_t, Y_{t-1}, ..., Y_{t-(m-1)}\} \tag{1}$$

where $m$ is the embedding dimension. In this case, with unit lags between time-series, $m$ corresponds to the length of the
segments and can be optimized using self-forecasting performance while predicting points in $Y_t$ from their nearest neighbors in $M_Y$ (Sugihara and May, 1990; Delforge et al., 2020a). This length $m$ is set to two days in this study due to the overall good performance of this value during our preliminary testing, i.e., $M_Y = \{Y_t, Y_{t-1}\}$.

For a causal analysis and to check if $X_t$ is a cause of $Y_t$, CCM makes forecasts of the points in $X_t$ from other values in $X_t$, whose time indices are identified from $Y_t$ as similar states in $M_Y$ for a given forecast time of reference. The algorithm is detailed
and illustrated in the supplementary materials (SM1.2). Our CCM implementation is found in Delforge et al. (2020a). CCM is an ensemble method, i.e., producing many ($N_{SAM}$) forecasts of the time-series at a given lag from $N_{SAM}$ bootstrapped samples of size $L$ from the trajectory matrix $M_Y$ to identify similar states. In our case, we measure the predictive skills as suggested in Sugihara et al. (2012) with the average Pearson's correlation $\bar{\rho}$ between time-series and their forecasts. Like CCF, the significance of the mean Pearson's correlation is appreciated with a Student's t-test.

There are few recent applications of CCM in hydrology (Wang et al., 2018; Medina et al., 2019; Ombadi et al., 2020). We chose the approach of Ye et al. (2015) over the original one of Sugihara et al. (2012). The latter infers causality without predictive delays ($d_{max} = 0$) and investigates convergence instead of time-dependencies. The convergence criterion for causality suggests that predictive skills $\bar{\rho}$ must increase by considering longer sample sizes $L$ in $M_Y$ to identify similar states until a plateau is reached. Yet, Sugihara et al. (2012) recognized that convergence could be met when variables are subject to strong
forcing or seasonality, a case referred to as synchrony, which could yield spurious bidirectional causal relationships, in particular regarding hydrological variables (e.g., Sugihara et al., 2017). Although the parallel is not explicit in the literature, we considered that synchrony relates to confounding and adopted the approach of Ye et al. (2015), which attempt to overcome synchrony by investigating the asymmetric patterns of time-dependencies and solve causality from the principle of priority up to $d_{max}$, i.e., as CCF but considering nonlinear dependencies (see also Wang et al., 2018). We do not check for convergence
and apply CCM with $L = 100$ based on our preliminary testing (similarly to Ombadi et al., 2020), assuming that convergence is occurring if $\bar{\rho}$ is significantly high for this given $L$. Secondly, we use a Theiler window (Theiler, 1986), $tw$, which is not considered in most applications of CCM (e.g., Wang et al., 2018; Ombadi et al., 2020), but well in applications related to the chaos theory from which CCM is derived (see Huffaker et al., 2018). The Theiler window is a time window that forces similar states to be sampled outside its range, for a time of reference. It prevents undesired predictive skills from being imputed to
auto-correlation rather than based on the consistency of dynamic recurrent trajectories, i.e., what CCM is supposed to test. Given the auto-correlated nature of hydrological series, we consider its use recommendable, especially for short time-series (Theiler, 1986). Although, the effect of $tw$ on predictive skills is expected to be marginal if the time-series are long enough (Delforge et al., 2020a). In our case, we systematically consider $tw = 10$ days such that similar states are most likely to be separated by one rainfall event.

Finally, since CCM is rooted in the chaos theory, it could be argued that CCM applies under the strict assumptions of its parent theory built on deterministic mathematical models, i.e., on low-dimensional systems with infinite length, noiseless, and non-cyclic and non-intermittent series or relations (Kantz and Schreiber, 2003; Sugihara et al., 2012; Sivakumar, 2017; Huffaker et al., 2018; Runge et al., 2019a). While we acknowledge that these hypotheses are not met with real hydrological time-series (Koutsoyiannis, 2006), CCM and its underlying framework were introduced as capable of operating on short and

noisy series as a nearest-neighbor regressor combined with its bootstrapping strategy (Sugihara and May, 1990; Sugihara et al., 1994, 2012), and the interest of any CIM is ultimately to be applied to real data while raising awareness on potential problems related to the real context. CCM, as well as other CIMs (see Runge et al., 2019a; Ombadi et al., 2020) or any statistical methods, meet issues with sample length, noise, cycles, dynamic intermittency, as well as the stationarity or significance testing (Huffaker et al., 2018; Medina et al., 2019), which should be considered in the result interpretation.

### 130   2.1.3   PCMCI: the Partial Correlation (ParCorr) and Conditional Mutual Information (CMI) tests

PCMCI is based on a stochastic framework testing conditional independence (Spirtes et al., 2000; Pearl, 2009), adapted to account for highly time-dependent time-series, and implemented in the Tigramite Python package (Runge et al., 2019b). It follows a two-step procedure: PC and Momentary Conditional Independence (MCI). The PC acronym refers to its authors Peter Spirtes and Clark Glymour (1991). PCMCI's grounding lies in a multivariate definition of the Wiener-Granger causality

(Wiener, 1956; Granger, 1969), here referred to as MCI (Runge et al., 2019b, a): a lagged variable $X_{t-d}$ is said to be a cause of $Y_t$ if $X_{t-d}$ has a significant dependence or predictive power over $Y_t$, while removing the effect of all other potential variables influencing $X_{t-d}$ or $Y_t$, except $X_{t-d}$. These potential variables are called Parents and symbolized $\mathcal{P}(X_{t-d})$ and $\mathcal{P}(Y_t) \setminus \{X_{t-d}\}$. Conditioning the analysis on the Parents allows dealing with confounding variables.

    Parents are subset variables found in the dataset that includes delayed variables up to $2d_{max}$, as $\mathcal{P}(X_{t-d})$ has to be estimated

beyond the maximum causal lag $d_{max}$. Parents are estimated during the conditions selection step performed with PC. The resulting estimates are noted $\hat{\mathcal{P}}(X_{t-d})$ and $\hat{\mathcal{P}}(Y_t) \setminus \{X_{t-d}\}$. PC removes irrelevant conditions from the full set of variables and delays by iterative independence testing until there is no more condition to drop out (see Runge et al., 2019b). The resulting size of Parents' set are controlled by $\alpha_{PC}$, a liberal parameter varying between 0 and 1, with the latter being the less restrictive case that includes all possible variables. Then, MCI is defined as:

$MCI : X_{t-d} \perp\!\!\!\perp Y_t | \hat{\mathcal{P}}(Y_t) \setminus \{X_{t-d}\}, \hat{\mathcal{P}}(X_{t-d})$                            (2)

To assert that $X_{t-d}$ causes $Y_t$, MCI must be rejected by a given independence test. PCMCI is paired with the linear Partial Correlation (ParCorr) and the nonlinear Conditional Mutual Information (CMI) tests. ParCorr is estimated by ordinary least squares by regressing the variables against their covariates considering a multivariate linear model. Then, the dependency between the residuals is tested using Pearson's correlation and a Student's t-test for the significance. For the PC step with ParCorr,

Tigramite optimizes $\alpha_{PC}$ between 0.05 and 0.5 based on Akaike's Information criterion (Akaike, 1974). In the information theory, CMI, or $I_{X,Y|Z}$ for possibly multivariate and continuous random variables $X, Y, Z$ is defined as the mutual information

$I$ between $X$ and $Y$ conditioned to $Z$ (Runge, 2018b):

$$I_{X,Y|Z} = \iiint dx\,dy\,dz\, p(x,y,z) \log \frac{p(x,y|z)}{p(x|z)p(y|z)} \tag{3}$$

If $I_{X,Y|Z} = 0$, $X$ and $Y$ are conditionally independent to $Z$, and, therefore, not directly causally related, assuming that the probability densities are correctly estimated, among other assumptions (Runge, 2018a). This is not a trivial task, especially in the case of high dimensionality or small sample sizes. Regarding PCMCI, dimensionality varies between variables according to the size of the sets of Parents (Eq.2), hence, depending on $d_{max}$, $\alpha_{PC}$, the independence test and its parameters. Based on numerical experiments covering sample sizes from 50 to 2,000 and dimensions up to 10, Runge (2018b) recommends using nearest-neighbors estimators of CMI (Frenzel and Pompe, 2007; Vejmelka and Paluš, 2008) for small sample sizes ($< 1000$). As no analytical significance test is available, like the Student's t-test used in the other cases, Runge (2018b) provides a shuffling significance test that is, in return, computationally demanding. Because of this computational requirement, PC with CMI has no implemented procedure for the selection of $\alpha_{PC}$ and we limited ourselves to two values among suggested ones: the default $\alpha_{PC} = 0.2$ and a more restrictive $\alpha_{PC} = 0.05$ given the strong interdependence of hydrological series.

PCMCI is based on a strict framework of assumptions: faithfulness, causal sufficiency, the absence of contemporaneous dependencies, the Causal Markov Condition, stationarity, and the assumptions behind the selected independence tests such as linearity or nonlinear constraints, or hyperparameters related to the estimators (see Runge, 2018a). In this study, we are mainly concerned with the first three. Faithfulness relates the absence of causality to conditional independence and imposes that time-series contains noise signals to test it. Causal sufficiency implies that the monitored variables include all common causes, following the principle (Reichenbach, 1956). Finally, contemporaneous links are related to the principle of priority, as some time lags are desired to infer the direction of causal relationships.

Given its flexibility, PCMCI is comparable to several other CIMs found in the literature. First, the PC algorithm itself is a CIM (Spirtes and Glymour, 1991), evaluated in hydrology as well (Ombadi et al., 2020). Yet, in comparison, the sequential PCMCI procedure is expected to reduce the number of False Positives (Runge et al., 2019b). PCMCI-ParCorr is also related to Granger Causality (GC) (Granger, 1969), which has several applications in hydrology (e.g., Salvucci et al., 2002; Tuttle and Salvucci, 2017), including karst (Kadić et al., 2018). Still, there are substantial differences. GC relies on multivariate vector autoregressive models, which do not account for contemporaneous dependencies. PCMCI reports significant contemporaneous dependencies, although without any causal direction since it cannot be inferred from the principle of priority. Also, GC relies on an F-test checking for significant reductions of the variance in the models' residuals. Besides, GC is not a sequential procedure like PCMCI. The GC conditioning is performed on all possible past variables, as if $\alpha_{PC} = 1$. As results, GC suffers from the curse of dimensionality, lowering its detection performance (Runge et al., 2019b, a). For this reason, GC is often applied in a bivariate way, considering only $X_t$ and $Y_t$'s past. The Transfer Entropy (TE) method (Schreiber, 2000) is the nonlinear and information-theoretic equivalent of bivariate GC (Barnett et al., 2009), making PCMCI-CMI a multivariate and sequential extension of TE. In hydrology, there are some applications of TE (e.g., Ruddell and Kumar, 2009; Sendrowski and Passalacqua, 2017; Rinderer et al., 2018; Ombadi et al., 2020). Regarding PCMCI-CMI, Goodwell et al. (2020) reviews

information-theoretic approaches and their potential applications in hydrology. Jiang and Kumar (2019) provides a framework built upon PCMCI-CMI to characterize short and long-term dependencies of observed stream chemistry data.

## 2.2 Study Site and Data

The karstic study site is the Lorette cave, next to the city of Rochefort in southern Belgium (Fig. 1.a) (Watlet et al., 2018; Poulain et al., 2018, and references therein). At 3 km of the study site, a PAMESEB agrometeorological station provides estimates of daily potential evapotranspiration data ($ET$, Fig. 1.c) with the Penman-Monteith FAO-56 method (Allen et al., 1998). On the Lorette site, at the Rochefort cave laboratory, daily rainfall data ($RF$, Fig. 1.a) are aggregated from a Lufft tipping bucket rain gauge with a 1 min sample rate located at the surface (elevation $\sim$225 m AOD). Inside the cave (elevation $\sim$190 m AOD), two drip discharge monitoring devices ($P1$, $P2$, Fig. 1.a) are installed within the main chamber accessible from a sinkhole, which constitutes the cave entrance. In particular, $P1$ monitors an active dripping point due to a visible fracture on the chamber's ceiling (Triantafyllou et al., 2019, for a 3D model). Based on dye injection at the surface and in-cave tracing, a connection and preferential flow path between the dye injection point (DT, Fig. 1.a) and $P1$ was identified (Poulain et al., 2018). The breakthrough curve showed an initial arrival time of 3.75 hours, a sustained peak for 80 hours, and a tail lasting up to 120 days. However, sporadic peaks in concentration were observed after every rainfall event, reacting after 1.48 hours, peaking after 7.2 hours, and lasting up to 30 hours on average. $P2$ monitors a dripping spot draining a porous limestone area. The last one, $P3$, located in the North gallery, monitors the slow discharge from drops falling from one single stalactite below a massive limestone layer. $P1$, $P2$, and $P3$ (Fig. 1.c) are daily means of the percolation rate.

An Electrical Resistivity Tomography (ERT) profile was installed to investigate the hydrology of the subsurface and potential connections above the cave (Watlet et al., 2018). ERT is a geophysical monitoring tool to study various types of hydrological processes (see Slater and Binley, 2021, and references therin). The ERT profile is not flat and starts from the depression of a sinkhole where the entrance to the cave is located (ERT, Fig. 1.a). The ERT experiment allowed collecting ERT data daily between 2014 and 2017, which represents the longest high-resolution ERT monitoring experiment conducted in a karst environment to our knowledge (see Watlet et al., 2018, for details). This dataset consists of processed ERT data, defined over 1558 spatial cells and 465 daily time-steps. As a necessary prior step to causal inference, the 1558 time-series were dimensionally reduced to six time-series clusters ($R0$ to $R5$, Fig. 1, b, and c). We used hierarchical agglomerative clustering with the Ward linkage method to minimize the squared distance between time-series within clusters. This algorithm is similar to k-means. As clustering was applied on the standardized ERT dataset, clusters represent groups of linearly correlated resistivity dynamics. The optimal number of clusters of 6 was selected using the optimal Silhouette Index as a clustering evaluation metric (Rousseeuw, 1987). The methodological aspects associated with clustering are extensively covered in another issue (Delforge et al., 2020b).

$R0$ is associated with a dense limestone area in the model's center (Fig. 1.b, X=22 m, Z=10 m). $R1$ covers responsive resistivity dynamics at the plateau's surface (Fig. 1.b, X=32 m, Z=13 m), and a fractured area around coordinate X=15 m below the surface pattern $R3$. $R2$ is rather representative of the resistivity patterns of the limestone matrix. $R4$ is associated

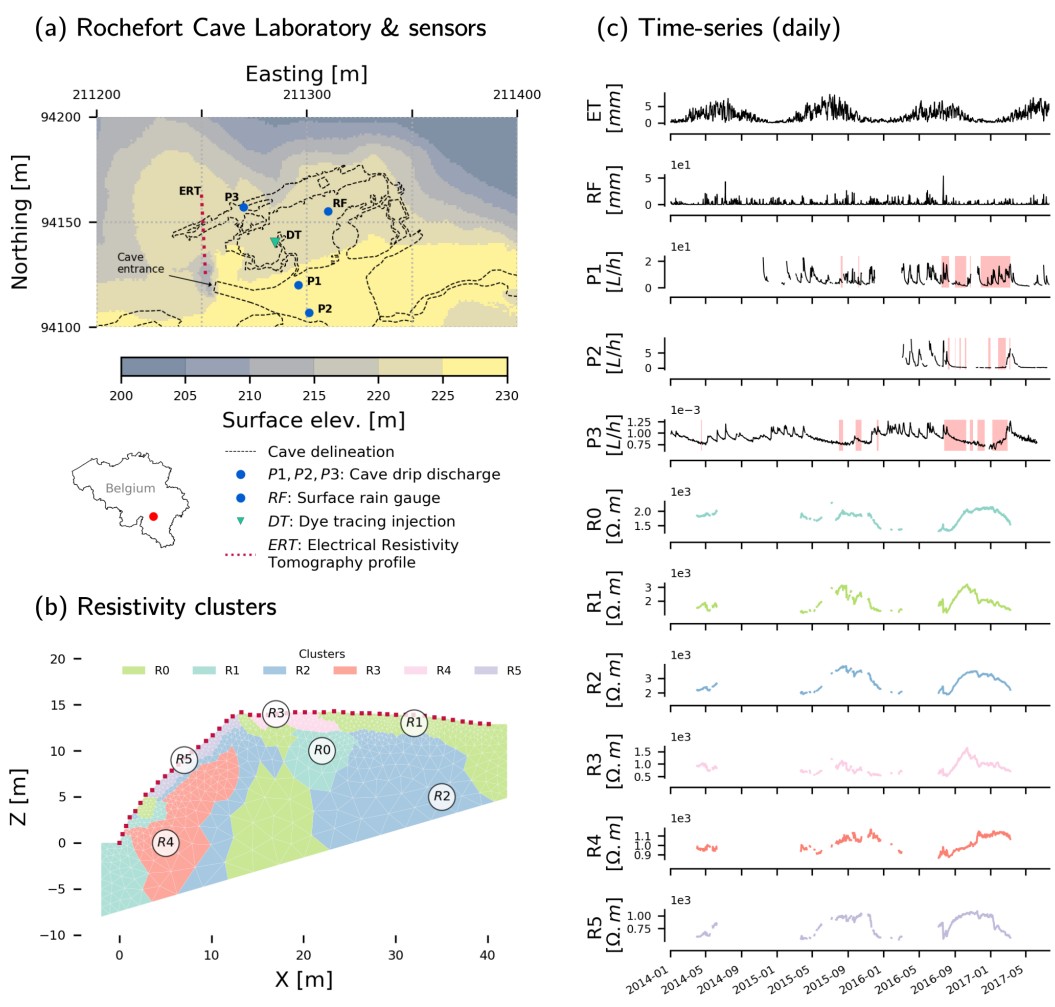

**Figure 1.** Study site and data: (a) Rochefort cave laboratory and sensors (EPSG: 31370), (b) resistivity clusters obtained from hierarchical agglomerative clustering of standardized resistivity data (Watlet et al., 2018; Delforge et al., 2020b), (c) daily time-series dataset. Resistivity time-series ($R0$ to $R5$) are the mean resistivity variations per cluster. Potential evapotranspiration data ($ET$) are obtained from an agrometeorological station (PAMESEB) located 3 km from the site. The red background areas in Fig. 1.c show the time-domain resulting from the conditioning on past delays with $d_{max} = 5$ days while considering respectively $P1$, $P2$, and $P3$ only in the causal dataset. A hydrological connection was identified by dye injection and tracing from the surface (DT) to $P1$ (Poulain et al., 2018). Source: Digital Elevation Model from Service Public de Wallonie, Cave delineation from Watlet et al. (2018).

**Table 1.** Summary statistics of the time-series variable.

| Statistic | ET [mm] | RF [mm] | P1 [L/h] | P2 [L/h] | P3 [L/h] | R0 [$\Omega$.m] | R1 [$\Omega$.m] | R2 [$\Omega$.m] | R3 [$\Omega$.m] | R4 [$\Omega$.m] | R5 [$\Omega$.m] |
|---|---|---|---|---|---|---|---|---|---|---|---|
| Count | 1297 | 1297 | 718 | 366 | 1223 | 465 | 465 | 465 | 465 | 465 | 465 |
| Mean | 2.2 | 2.0 | 6.13 | 1.30 | 9.06E-04 | 1.80E+03 | 1.95E+03 | 2.80E+03 | 8.78E+02 | 1.02E+03 | 8.28E+02 |
| Std dev. | 1.8 | 4.2 | 4.33 | 1.92 | 1.11E-04 | 2.38E+02 | 5.68E+02 | 5.64E+02 | 2.35E+02 | 7.75E+01 | 1.73E+02 |
| Min | 0.0 | 0.0 | 1.05 | 0.00 | 6.35E-04 | 1.30E+03 | 1.11E+03 | 1.88E+03 | 5.07E+02 | 8.67E+02 | 5.33E+02 |
| 10% | 0.3 | 0.0 | 1.91 | 0.05 | 7.76E-04 | 1.37E+03 | 1.28E+03 | 2.03E+03 | 5.92E+02 | 9.23E+02 | 5.89E+02 |
| 25% | 0.7 | 0.0 | 3.00 | 0.09 | 8.21E-04 | 1.69E+03 | 1.48E+03 | 2.27E+03 | 7.06E+02 | 9.59E+02 | 6.45E+02 |
| 50% | 1.6 | 0.1 | 4.57 | 0.27 | 8.99E-04 | 1.85E+03 | 1.85E+03 | 2.83E+03 | 8.51E+02 | 1.03E+03 | 8.52E+02 |
| 75% | 3.3 | 2.1 | 8.46 | 1.73 | 9.75E-04 | 1.96E+03 | 2.36E+03 | 3.29E+03 | 1.01E+03 | 1.10E+03 | 9.92E+02 |
| 90% | 5.0 | 6.2 | 12.74 | 4.20 | 1.06E-03 | 2.08E+03 | 2.86E+03 | 3.48E+03 | 1.21E+03 | 1.13E+03 | 1.03E+03 |
| Max | 8.6 | 53.8 | 22.66 | 9.45 | 1.26E-03 | 2.30E+03 | 3.21E+03 | 3.83E+03 | 1.65E+03 | 1.17E+03 | 1.09E+03 |

with a clayey limestone layer located below the pattern of slope surface $R5$ (Watlet et al., 2018; Delforge et al., 2020b). We expect causal links to appear primarily between P1 and the near-surface resistivity patterns $R1$ or $R3$.

Time-series of Fig. 1.c are the 11 inputs for the four selected CIMs. Table 1 shows their statistics. Bivariate CIMs (CCF and CCM) are applied between each pair of time-series on their overlapping time-domain with a maximum causal delay $d_{max} = 5$ days. We considered this delay for our final results as it covers the span of bivariate dependencies (Fig. SM2.1, SM2.2), and the full time-span of preferential flow peaks lasting up to 80 h (Poulain et al., 2018). For the same reason, we use the same $d_{max}$ for the multivariate methods. In general, large $d_{max}$ values are recommended to satisfy the hypothesis of causal sufficiency while PCMCI sequential procedure deal with the potentially high dimension resulting from $d_{max}$ (section 2.1.3). Still, $d_{max}$ cannot be higher than 5 days because of our uneven time distribution of missing data. Indeed, PCMCI dismisses all time slices of samples where missing values occur in any variable and their lags up to $2d_{max}$, which limits the overall overlapping time-domain to 48 days, mostly because of the short time-domain of $P2$ (Fig. 1.c, Table 1). Concerned that this number would impact the robustness of the analysis, we also applied PCMCI while considering one percolation data at a time. In this way, $P1$, $P2$, $P3$ are considered separate and independent drainage systems, and the overlap domains become larger while keeping $d_{max} = 5$ days: 184, 62, and 218 days respectively. These domains are shown in red for $P1$,, $P2$, $P3$ in Fig. 1.c.

## 3 Synthetic Case Study

### 3.1 Conceptual Model

We aim at assessing CIMs performances for the detection of effective hydrological connectivity, in particular, the assertion that multivariate nonlinear methods are best suited for that purpose. To this end, we built a simple hydrological reservoir model, inspired by the common cause problem (Fig. 2). Two separate and independent reservoirs, $A$ and $B$, and their discharge $Q_A$, $Q_B$, are forced with the same effective precipitation $P_{eff}$ (i.e., precipitation minus evapotranspiration). Then, without

any effective connection, they will nevertheless show a temporal dependence. If $A$ and $B$ are disconnected, the ideal CIM would then reject the effective connection between $Q_A$ and $Q_B$ if $P_{eff}$ is included in the multivariate analysis. However, $B$ responds systematically one day later than $A$ to $P_{eff}$. Hence, with bivariate methods and the priority principle only, $Q_A$ would seemingly cause $Q_B$. For comparison, we consider a case where $Q_A$ is effectively connected to another series $Q'_B$, as they were contributing to the same drainage network, with $Q_A$ upstream of $Q'_B$. Noteworthily, this experiment does not cover the case of nonlinearities arising from threshold effects and intermittent processes. Instead, we assume sustained causal interactions and no intermittency over time, as the CIMs do.

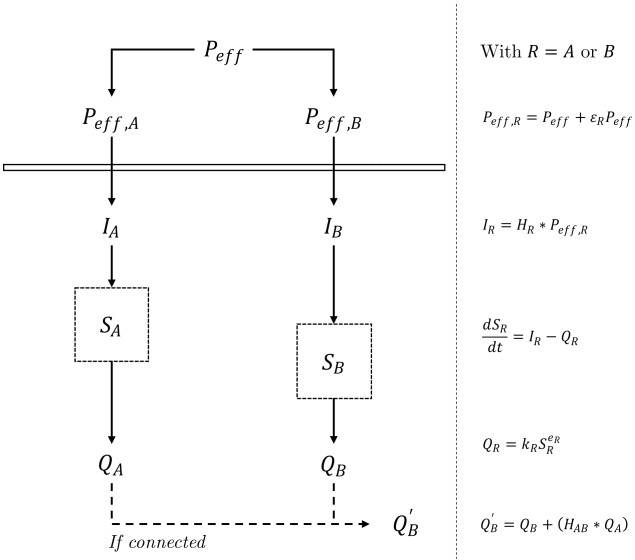

**Figure 2.** The conceptual and mathematical model for the synthetic case study. Two reservoirs $A$ and $B$ are forced by inflows $I_A$ or $I_B$ resulting from the unit hydrograph $H_A$ and $H_B$ forward convolution on noisy variant $P_{eff,A}$ and $P_{eff,B}$ of the same effective precipitation $P_{eff}$ computed from real data (section 2.2). The storage $S_A$ and $S_B$ dynamics follows a typical continuity equation $dS/dt = I - Q$, where the discharge $Q_A$ and $Q_B$ follow a nonlinear power law $Q = kS^e$ with two parameters $k$ and $e$. The reservoir $B$ responds mainly 1 day after $A$, which introduced a systematic time-dependency between the reservoir suggesting causation. $Q'_B$ is a flow downstream of $Q_B$ draining $Q_A$ and transferred considering the unit hydrograph $H_{AB}$. The causal analysis involves either the disconnected case $Q_A$ and $Q_B$ or the connected case $Q_A$ and $Q'_B$. Multivariate methods includes $P_{eff}$ within the analysis.

The model is forced by real effective precipitation data $P_{eff}$ monitored at the study site (section 2.2). Both reservoirs take as input a net inflow term $I_A$ and $I_B$ resulting from the application of the unit hydrographs $H_R$, with $R$ either $A$ or $B$, as linear transfer functions convolved forwardly on a noisy precipitation input ($H_R * P_{eff,R}$ with $*$ the convolution operator). Adding some noise is mandatory to check for conditional independence (faithfulness, section 2.1.3). A multiplicative noise term is preferred as hydrological variables are often characterized by multiplicative noise (e.g., Rodriguez-Iturbe et al., 1991). Accordingly, $P_{eff,R} = P_{eff} + \varepsilon_R P_{eff}$, with $\varepsilon_R$ being randomly generated from a normal distribution with zero mean and standard deviation equal to $\varepsilon_{lvl}$ times the standard deviation of $P_{eff}$. The parameter $\varepsilon_{lvl}$ is always identical for $A$ and $B$, such

that $\varepsilon_A$ and $\varepsilon_B$ have the same distribution. The continuity equation gives the reservoirs storage dynamics: $dS_R/dt = I_R - Q_R$. The outflow $Q_R$ introduces some nonlinearities through a typical nonlinear storage-discharge relationship $Q_R = k_R S_R^{e_R}$, with $k_R$ and $e_R$ the discharge coefficient and the nonlinear exponent. Such power-law formulations are typical in hydrology (Dooge, 1973) and common while modeling karsts as well (Hartmann et al., 2014; Jourde et al., 2015).

**Table 2.** Model parameters for the synthetic cases

| Model | $H_A$ | $k_A$ | $e_A$ | $H_B$ | $k_B$ | $e_B$ |
|-------|-------|-------|-------|-------|-------|-------|
| 1 | [0.7, 0.2, 0.1] | 0.1 | 1 | [0.1, 0.8, 0.1] | 0.1 | 1 |
| 2 | [0.7, 0.2, 0.1] | 0.1 | 1 | [0.1, 0.8, 0.1] | 0.01 | 1.5 |
| 3 | [0.7, 0.2, 0.1] | 0.01 | 1.5 | [0.1, 0.8, 0.1] | 0.1 | 1 |
| 4 | [0.7, 0.2, 0.1] | 0.01 | 1.5 | [0.1, 0.8, 0.1] | 0.01 | 1.5 |

For the synthetic cases, we derived four models based on Table 2. The unit hydrographs $H_A$ and $H_B$ are constant, with their maxima differing by one daily time-step. This lag introduces the desired constant time-dependencies between the two reservoirs despite the absence of connection. The recession parameters allow generating distinct dynamic patterns with various degrees of nonlinearity thanks to $e_R$. In addition, we also considered 14 stochastic noise level $\varepsilon_{lvl} \in \{0.05, 0.1, \ldots, 0.65, 0.70\}$. With such noise levels, the correlation between two generated $Q_A$ with model configuration 1 (Table 2) should vary on average between 0.96 ($\varepsilon_{lvl} = 0.05$) and 0.24 ($\varepsilon_{lvl} = 0.7$). With the four combinations of Table 2 and the 14 noise levels, 56 datasets were generated from four years of effective precipitation data $P_{eff}$ (2014-2018) and initial storages $S_A$ and $S_B$ equal to 30 mm. Only the last year of the three variables $P_{eff}, Q_A, Q_B$ were considered for the causal inference experiment, the first three being considered as a warming-up period. The data generation is repeated to produce 56 additional datasets with an effective connection. $Q_A$ and $Q_B$ are causally related by overwriting $Q_B$ such that $Q'_B = Q_B + (H_{AB} * Q_A)$ where $H_{AB} = [0.1, 0.8, 0.1]$ is a linear transfer function convolved forwardly on $Q_A$. Finally, the whole synthesis process is repeated to generate first-order differenced datasets (i.e., $Y_t - Y_{t-1}$), that is a total of 224 datasets with 4 model combinations, 14 noise levels, connected or not, and differenced or not. The primary purpose of the differenced data is to simply illustrate the effect and value of removing past dependencies (auto-correlation, seasonality) on the CIMs results.

## 3.2 Results

Figure 3 depicts the average and interquartile range of $Q_A$-$Q_B$ time-dependencies, or $Q_A$-$Q'_B$ if connected, obtained with the four CIMs (a to d) on the synthetic datasets for a maximum delay $d_{max} = 5$ days. The multivariate analysis includes $P_{eff}$. We distinguish between cases where the reservoirs are connected or not and where the data are differenced or not. Regarding the bivariate methods (a and b), CCF and CCM both exhibit sustained time-dependencies for not-differenced data due to the auto-correlation and seasonal patterns left in the series. For differenced data, the results better screen the expected peak at lag one, including disconnected reservoirs. This illustrates that bivariate CIMs cannot deal with the confounding effect related to the common forcing of reservoirs. This is not an effective connection but a functional and apparent one resulting from their

delayed responses. The sustained time-dependency of CCM over the lag of 2 days is an artifact of the embedding dimension, $m = 2$ (Eq.1), defining the length of trajectory segments, which is two days in this case (see SM1.2).

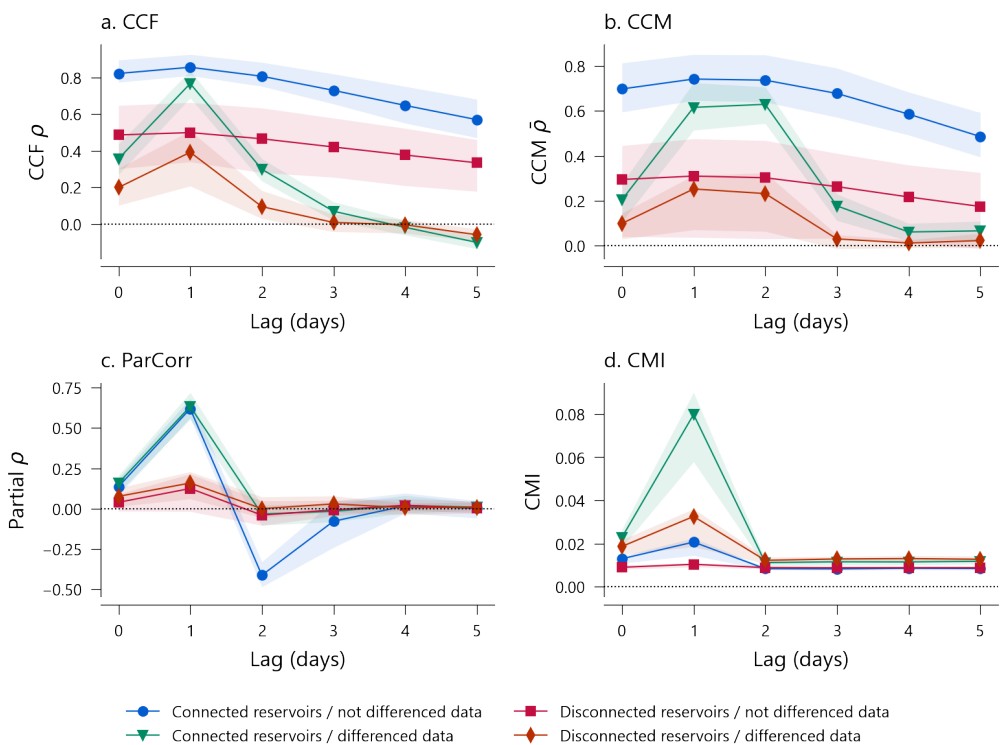

**Figure 3.** Patterns of statistical time-dependencies between $Q_A$ and $Q_b$ ($Q'_B$ if connected) for the four CIMs (a. to d.). Lines are the average statistics for 56 synthetic datasets obtained from four different model structures (Table 3) and 14 distinct noise levels. The envelope represents the interquartile range of the statistics. In general, connected reservoirs show a $Q_A \rightarrow Q'_B$ causal dependencies at a lag of one day. However, except for CMI on the not differenced data, disconnected reservoirs show a non-causal, yet, significant statistical dependencies $Q_A \rightarrow Q_B$ at lag one day, since reservoir $A$ mostly reacts one day before $B$ to the effective precipitation.

Regarding the multivariate CIMs (c and d), differencing time-series is theoretically useless as PCMCI already conditions on the past variables (Eq. 2) and can deal with auto-correlation and seasonality. The linear ParCorr method seems to discriminate the connected case from the disconnected case. Still, it always shows a peak at lag 1, whatever the cases, which could be misinterpreted as an effective connection. Only the nonlinear CMI method applied to the not-differenced data seems to reject the idea of connection when it is effectively absent. This finding supports our theoretical assertion: the multivariate nonlinear method is best suited to address effective hydrological connectivity. Furthermore, the method appears to perform better if seasonality is left present in the time-series. Still, Fig. 3 shows the pattern of the statistics, not the result of a causality test and its p-value.

**Table 3.** Causal test statistics for the synthetic cases at lag 1 day

| Method | ParCorr | | CMI $\alpha_{PC} = 0.05$ | | CMI $\alpha_{PC} = 0.2$ | |
|---|---|---|---|---|---|---|
| **Datasets** | Not diff. | Diff. | Not diff. | Diff. | Not diff. | Diff. |
| **Data Statistics** | | | | | | |
| Total count | 112 | 112 | 112 | 112 | 112 | 112 |
| Actual + | 56 | 56 | 56 | 56 | 56 | 56 |
| Actual - | 56 | 56 | 56 | 56 | 56 | 56 |
| **Test results** | | | | | | |
| TP at 99% | 56 | 56 | 28 | 56 | 15 | 56 |
| (95%) | (56) | (56) | (34) | (56) | (27) | (56) |
| FP at 99% | 29 | 29 | 3 | 28 | 1 | 28 |
| (95%) | (35) | (32) | (4) | (42) | (2) | (35) |
| TN at 99% | 27 | 27 | 53 | 28 | 55 | 28 |
| (95%) | (21) | (24) | (52) | (14) | (54) | (21) |
| FN at 99% | 0 | 0 | 28 | 0 | 39 | 0 |
| (95%) | (0) | (0) | (22) | (0) | (29) | (0) |
| **Test metrics** | | | | | | |
| Accuracy[1] 99% | 0.74 | 0.74 | 0.72 | 0.75 | 0.64 | 0.75 |
| (95%) | (0.69) | (0.71) | (0.77) | (0.62) | (0.72) | (0.69) |
| Precision[2] 99% | 0.66 | 0.66 | 0.90 | 0.67 | 0.94 | 0.67 |
| (95%) | (0.62) | (0.64) | (0.89) | (0.57) | (0.93) | (0.62) |
| Recall[3] 99% | 1.00 | 1.00 | 0.5 | 1.00 | 0.3 | 1.00 |
| (95%) | (1.00) | (1.00) | (0.61) | (1.00) | (0.48) | (1.00) |
| FP rate[4] 99% | 0.52 | 0.52 | 0.05 | 0.5 | 0.02 | 0.5 |
| (95%) | (0.63) | (0.57) | (0.07) | (0.75) | (0.04) | (0.63) |

[1](TP+TN)/(TP+FP+FN+TN);   [2]TP/(TP+FP);   [3]TP/(TP+FN);   [4]FP/(FP+TN)

TP: True Positive; FP: False Positive; TN: True Negative; FN: False Negative

Since we know for each simulation whether or not there is a causal link between $A$ and $B$, Table 3 reports True Positives (TP), False Positives (FP), True Negatives (TN), and False Negatives (FN) for the problematic lag of 1 day. We consider the multivariate PCMCI methods ParCorr and CMI, with the latter having two different $\alpha_{PC}$ for preselection of Parents (PC stage). Two levels of significance are considered at 99% and 95% based on the p-values obtained by the tests. Table 3 shows that, for a similar level of accuracy, CMI for not-differenced data has higher precision and a lower false-positive rate, meaning that positive tests are likely to detect actual causal relations. Differencing increased the PCMCI-CMI false-positive rate. The low false-positive rate for not-differenced data is particularly contrasting with other methods and provides a valuable piece of information. However, the high precision comes at the cost of a low recall (as reported in Runge, 2020): CMI misses about half of the actual causal links. On the contrary, ParCorr misses none but has a bad precision, i.e., many False Positives. This analysis thus provides an overview of the contrasts between methods. Of course, this virtual three-variable configuration is far from representative of the great variety of natural hydrological systems and their spatiotemporal organizations.

## 4 Real Case Study

### 4.1 Bivariate Methods

Figure 4 reports the causal graphs for significant pairwise dependencies between first-order differenced time-series, for a better screening of time-dependencies using bivariate methods (Fig. 3). Detailed time-dependencies are also reported in the Supplementary Materials (SM2.1, SM2.2), allowing us to consider that $d_{max} = 5$ days is sufficient. The CCF method (Fig. 4.a) reports many potential linear causal associations (arrows), in red for positive correlations and in blue for negative ones. Delayed dependencies are represented with curved arrows with the delay $d$ (in days) printed in the middle. Contemporaneous dependencies ($d$=0) are represented by straight links, with no direction since it cannot be inferred from the principle of priority. For CCM (Fig. 4.b), significant dependencies are fewer, unsigned as nonlinear, and, therefore, only represented in red. The results and their meaningfulness are appreciated in their dedicated discussion sections.

### 4.2 Multivariate methods

For multivariate methods, we chose to report causal graphs for the raw (not-differenced) data since differencing is theoretically unnecessary and reduced the precision of CMI in the virtual experiment (Table 3). Hence, Fig. 5 shows linear conditional dependencies (ParCorr) obtained from the raw time-series for the full dataset (All data, Fig. 5.a) and considering the discharge series one by one (P1, P2, and P3, Fig. 5, b to d). The P1, P2, and P3 datasets allow the analysis to be performed over larger time domains (Fig. 1.c). Except for R4, the dominant relationships between resistivity and meteorological variables are maintained between the graphs, demonstrating stability in the ParCorr results despite differences in the considered time-domain.

Regarding PCMCI-CMI, we found unstable results on all datasets: All data, P1, P2, and P3 (see SM2.3). The causal graphs varied substantially when we repeated the analysis with the same parameters due to the stochastic nature of the independence test (Runge, 2018b). Consequently, we developed a sensitivity analysis by varying the hyperparameters of the method, hoping

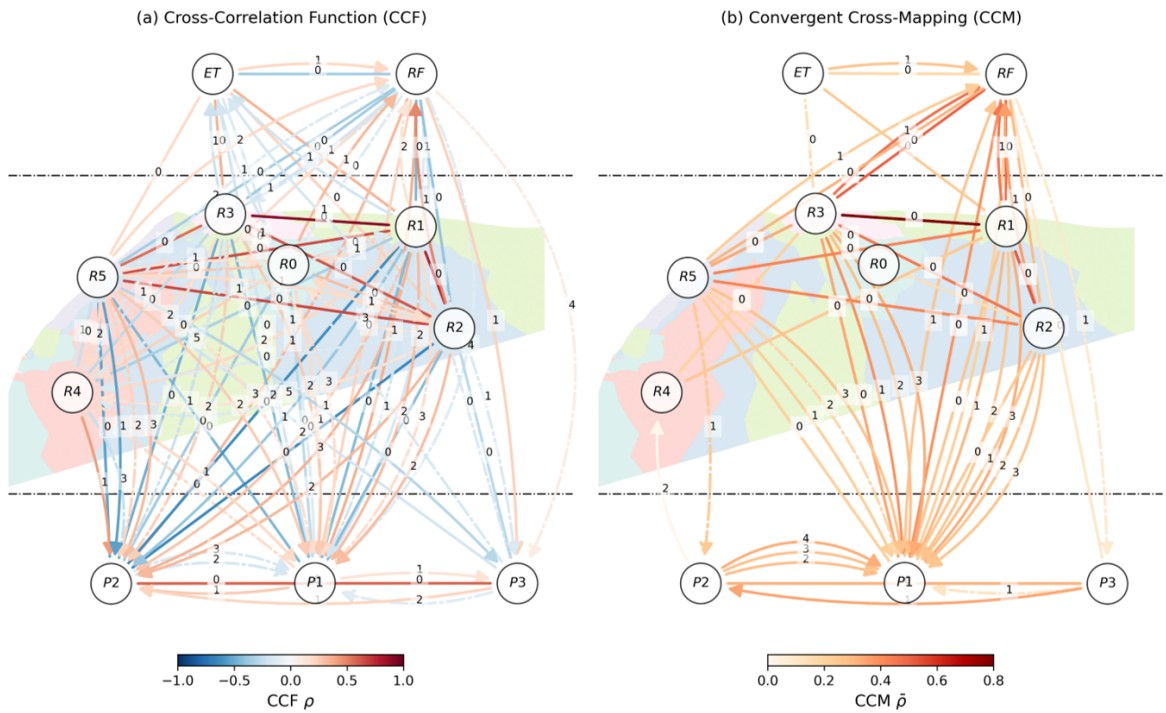

**Figure 4.** Graph of pairwise cross-dependencies: (a) with the linear Cross-Correlation Function (CCF),(b) with the nonlinear Convergent Cross-Mapping (CCM) method. An undirected line represents contemporaneous dependencies. Delayed dependencies are shown using directed curved arrows. All corresponding delays $d$ are displayed in the middle of its corresponding arrow. The color of arrows maps to the strength of dependencies. Solid and dash-dotted arrows represent respectively significant dependencies with p-value < 0.001 and < 0.01.

to isolate more stable configurations (SM2.3). Whatever the configuration, the results remained unstable. Consequently, to
achieve a causal representation of the system with the nonlinear multivariate method, we had to adopt another logic, that of the
consensus brought by the set of models from the sensitivity analysis. We considered all simulations done for the sensitive analysis as an ensemble of models rendering each a causal graph. Figure 6 reports the links achieving majority (50% considering a
p-value of 0.05) among the ensemble of causal graphs from the sensitivity analysis.

## 5   Discussion

### 5.1   CIMs' specific discussion

#### 5.1.1   Cross-Correlation Function (CCF)

The CCF results for the real case, on the differenced data, show a dubiously high number of significant connections; we interpret
those as an indication that the method does not only reveal effective hydrological connections and reflects the complexity

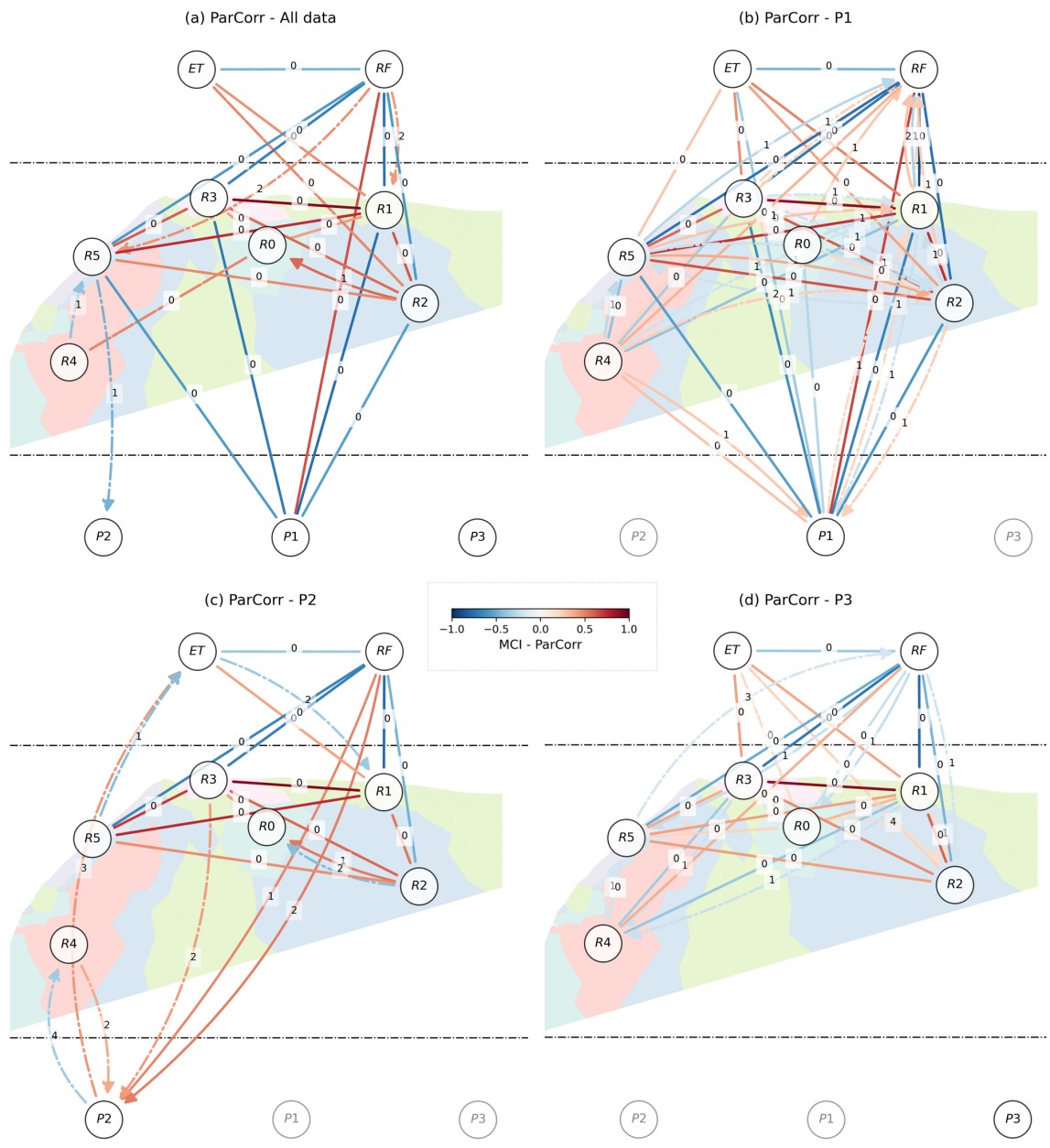

**Figure 5.** Graph of ParCorr cross-dependencies: considering (a) all data or one unique discharge series (b) P1, (c) P2, or (d) P3. An undirected line represents contemporaneous dependencies. Delayed dependencies are shown using directed curved arrows. All corresponding delays $d$ are displayed in the middle of its corresponding arrow. The color of arrows maps to the strength of dependencies. Solid and dash-dotted arrows represent respectively significant dependencies with p-value < 0.001 and < 0.01. For each graph, the size of the overlapping time-domain between the variables changes as follows: 48 days (a), 184 days (b), 62 days (c), and 218 days (d).

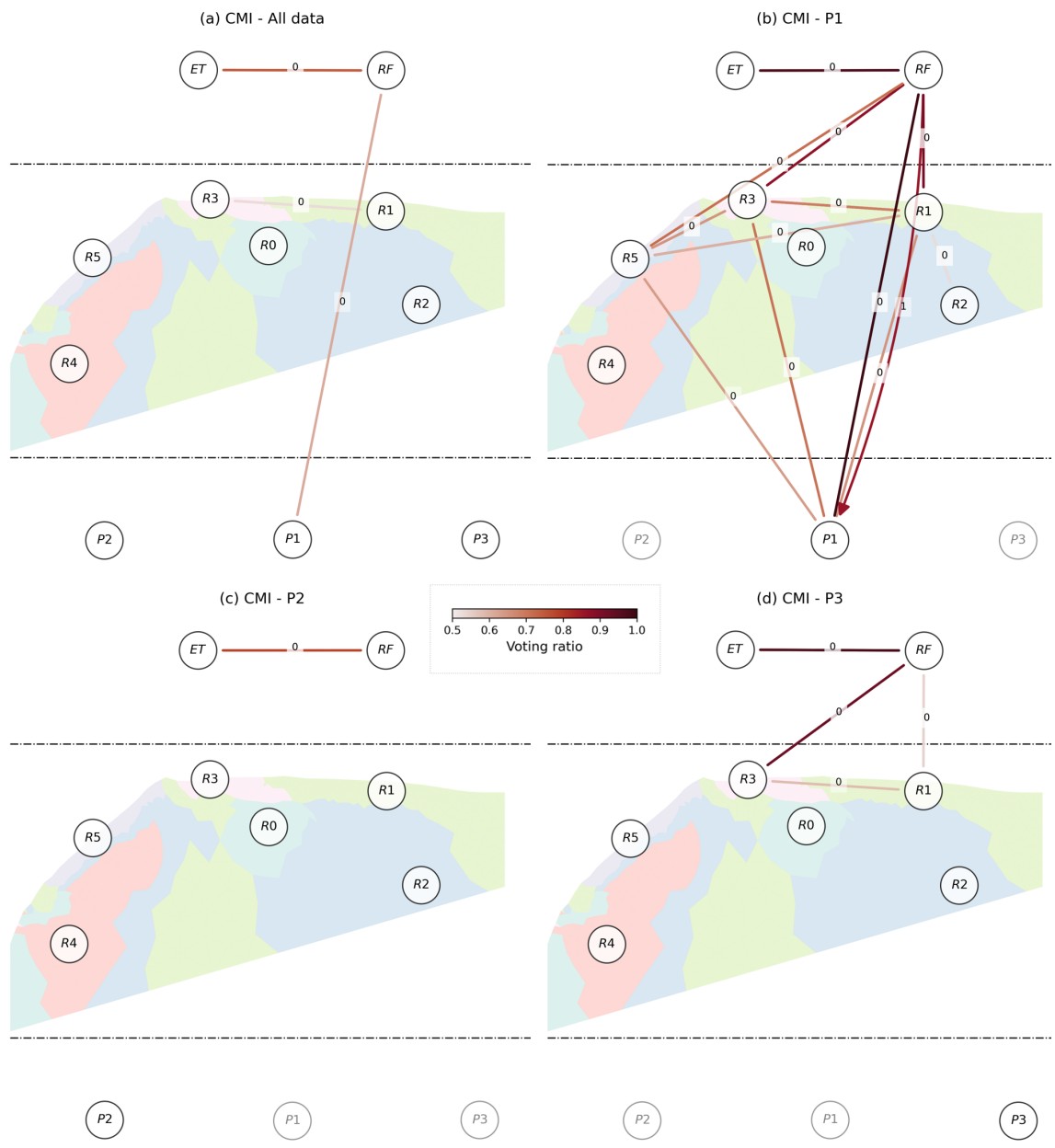

**Figure 6.** Consensual graph of CMI cross-dependencies obtained from the ensemble of simulations performed in the sensitivity analysis: (considering (a) all data or one unique discharge series (b) P1, (c) P2, or (d) P3. An undirected line represents the Contemporaneous dependencies. Delayed dependencies are shown using directed curved arrows. All corresponding delays d are displayed in the middle of its corresponding arrow. The links displayed are those reaching the majority (50%) for all causal graphs where the connections are established with a p-value of significance at 0.05. The color of arrows maps to the strength of the voting ratio. For each graph, the size of the overlapping time-domain between the variables changes as follows: 48 days (a), 184 days (b), 62 days (c), and 218 days (d).

of inferring connectivity from CIMs with strongly correlated, synchronous series forced by common drivers. As a result, statistical dependencies, or functional connections, are ubiquitous. A corollary would be: one could commonly build a point-to-point model based on two time-series and their statistical dependencies (functional connections) in hydrology. Somehow, it explains why hydrological systems may be modelled with simple functional lumped model or linear transfer functions (Dooge, 1973). However, when effective connections matter, e.g., to predict the spread of a pollutant, it might become problematic. Similarly, Rinderer et al. (2018) reported many connections with the CCF method and bivariate Granger Causality. Our primary explanation is that CCF on differenced data, i.e., with seasonality roughly removed, does not account for the common forcing by rainfall, evapotranspiration, or residual harmonic signals. This failure leads to dependencies despite the absence of causal links or effective connections (as in Fig. 3a), and the system's variables remain highly correlated and synchronized.

Besides, we see that our results contain many False Positives reasonably identified from the sign of the dependencies (Fig. 4a). The fact that linear methods report signed dependencies makes it easier to interpret the results and potential False Positives. For instance, positive relationships between resistivity and drip discharge are unexpected if water transfers from low resistivity areas to the drip outlets. Often, positive relationships, e.g., $R5 \rightarrow P2$ with $d = \{2, 3\}$, follow negative ones that are more interpretable as a transfer (e.g., $R5 \rightarrow P2$ with $d = \{0, 1\}$). This pattern is rather a phasing artifact captured by CCF interpretable as: "after the rain, the good weather", and vice-versa. In general, we observe strong linear dependencies between resistivity series (except $R4$, the clayey limestone), which may be a problem for PCMCI-CMI (see 5.1.4). Some links are more intriguing and are hardly interpreted as a causal relationship, e.g., $R5$ causing $RF$.

From this latter observation, we recall that dependencies can appear by chance, in particular, if the series are short, hence the importance of significance tests. As suggested in Rinderer et al. (2018), we could have further controlled the number of links by considering lower p-values, but these were already as low as 0.001, or by comparing the dependencies obtained with those obtained with surrogates (Ebisuzaki, 1997). Surrogates are random substitutes for the original data that share statistical or dynamic properties with the original data, for example, auto-correlation patterns (Schreiber and Schmitz, 2000). In our opinion, these approaches are convenient to rule out many connections that prevent us from focusing on the most significant ones. They will, however, not overcome the method drawback that is imputed to its design. As a bivariate method, CCF may fail to retrieve effective connections due to confounding factors (Runge et al., 2019a, and section 3.2).

Though clearly not perfect, CCF is simple, linearly interpretable, and widespread or popular. Considering that other CIMs carry their own imperfection, we do not discourage using CCF while knowing its limitations, testing for linearity, removing harmonics (or the use of suitable surrogates' comparisons), or confounding factors using either a statistical or physical model. Manual handling of confounding factors is close to assessing connectivity/causality in a multivariate framework, although not automated but supervised by our knowledge and expertise. Here, we only removed the seasonal signal by differencing, which is not sufficient and results in many dependencies.

### 5.1.2 Convergent Cross Mapping (CCM)

CCM results present fewer connections (Fig. 4.b), with many connections pointing from $RF$ or resistivity series (except for $R4$, the clayey limestone) to $P1$. In particular, this result is encouraging as we expect a fast preferential flow and effective

connection between the surface and $P1$ (Poulain et al., 2018, and Fig. 1). Still, even when applied to deterministic time-series (Runge et al., 2019a), for a system not subject to synchrony (Sugihara et al., 2012, 2017), or, in such case, by using the time-delayed approach of Ye et al. (2015), CCM can be deceiving because it cannot deal with confounding factors, by design, as a bivariate method. Our result in the synthetic experiment tends to confirm it (Fig. 3.b). CCM identifies sustained significant dependencies with the not-differenced data imputed to slow seasonal variations. As for CCF, some of these dependencies may have been ruled out if we had considered surrogate significance testing with a surrogate model that accounts for seasonality (as seen in Nes et al., 2015; Huffaker et al., 2018; Medina et al., 2019). Yet, on the differenced data removing most of the seasonal signal, CCM captures significant time-dependencies at delays reflecting the response time of the reservoirs and not effective connections since they are disconnected. Besides, for the real experiment (Fig. 4.b), we found dependencies between drip discharge data (e.g., $P2$ and $P1$), which cannot be interpreted as effective connections, since they represent different outlets, similarly to the virtual experiment. In parallel, Ombadi et al. (2020) reported a high false-positive rate while using the original CCM approach (Sugihara et al., 2012), whatsoever the sample length, explained by CCM's inability to deal with confounding and synchrony.

In addition, noise is expected to reduce both the CCM's true and the false-positive rates, i.e., the general mapping skills (see Ombadi et al., 2020). Regarding our experiment as well, although hidden in the interquartile range shown in Fig. 3b., we found that higher noise levels led to lower $\bar{\rho}$. Theoretically, however, this decrease in performance might not be systematic. It may, in some cases, depend on the auto-correlated nature of noise and deterministic signals, the sample length, and the size of a Theiler window (Theiler, 1986). In general, we consider the idea of CCM being restricted to strictly deterministic systems (Runge et al., 2019a) to be overly conservative, potentially misleading, and impractical to real case studies. The theory does not impose anything on the practitioners, but it raises awareness of potential problems if the assumptions are not tested or violated, which is why it deserves careful considerations (further developed in Kantz and Schreiber, 2003; Sivakumar, 2017; Huffaker et al., 2018).

Currently, we consider CCM to be best suited to test whether or not there is a functional connection between two points, similarly to CCF but considering nonlinear dependencies. Therefore, our recommendations align with those of CCF. However, if CCM has better predictive capabilities than CCF, it can be concluded that the dependency is nonlinear.

### 5.1.3 PCMCI: Partial Correlation (ParCorr)

ParCorr is linearly interpretable and computationally efficient. As CCM, the ParCorr results for the real experiment (Fig. 5) seem promising because they generally favor the expected preferential flow and effective connection between the surface and $P1$ (Poulain et al., 2018, and Figure 1). ParCorr did not experience any stability problem, first, because ParCorr always provides the same results for the same dataset. Secondly, we deemed that the most significant connections remain between the results from the dataset variations that we tested (Fig. 5.a-d). Although there are few differences between the links that are evaluated in the four variations, these are often related to relationships of lesser significance or magnitude. Some of them are alarming, such as $R5 \rightarrow RF$ at lag $d = 1$ in Fig. 5.b or $d = 3$ in Fig. 5.d. They are even more alarming because the whole causal graph can be affected by conditioning on an irrelevant variable (Eq. 2).

Based on the results of the synthetic experiment (Fig. 3 and Table 3), PCMCI-ParCorr, or its variant multivariate GC, may suffer from a relatively high false-positive rate (> 50% in Table 3), most likely due to the inability to deal with direct nonlinear dependencies. Similarly, Ombadi et al. (2020) reported a high false-positive rate for GC in their synthetic experiment, although below CCM's one. A high rate was also observed in Rinderer et al. (2018), although using bivariate GC. Accordingly, claiming an effective hydrological connection based on PCMCI-ParCorr requires caution because of the high false-positive rate reported here and in the literature. We consider that its applications would require deeper testing to ensure that a multivariate linear model fits the data.

### 5.1.4 PCMCI: Conditional Mutual Information (CMI)

The result of the virtual experiment (Fig. 3 and Table 3) confirmed our general expectation that a multivariate nonlinear method is best suited to assess effective connectivity. PCMCI-CMI had the lowest false-positive rate (Table 3), which is particularly desired given the confounding problem in hydrological systems. Yet, it had a low recall relative to other approaches, meaning that the results contain False Negatives. Ombadi et al. (2020) also reported a relatively low false-positive rate using different methods, PC (Spirtes and Glymour, 1991) and the bivariate conditional Transfer Entropy (TE) method (Schreiber, 2000). Still, according to Runge et al. (2019b), PCMCI-CMI is supposed to perform better than PC thanks to its sequential procedure and TE by being multivariate and accounting for confounding effects.

In the real case study, PCMCI-CMI was found unstable, providing different results across consecutive runs (see, Fig SM.5 and SM.6).Two main reasons might explain this instability. First, PCMCI conditions dependencies on the Parents (Eq. 2) and, therefore, build a dataset containing the initial variables and their delayed versions up to $2d_{max}$. The overlapping time-domain over which all variables and delays are defined is small (see Fig. 1). It covers only 48 up to 218 time-stamps (P3). Ruddell and Kumar (2009) reported that 500 to 1000 samples are generally sufficient to obtain a qualitatively robust estimate of the TE, estimated using a binning approach. The nearest-neighbor approach that we used (Runge, 2018b) is more suited for short datasets. Yet, we found that our final sample sizes, accounting for missing data, are lower and the dimensionality is potentially higher, though variable and depending on the size of the Parents' sets (Eq.2), than those tested in the evaluation of the CMI estimator (Runge, 2018b). This could be the main reason for the instability. In addition, we applied our analysis to highly correlated data from a smooth inverted electrical resistivity model (Fig. 4.a). PCMCI-CMI encountered highly interdependent anomalies regarding resistivity (reported in Fig. SM.5 and 6) . We suspect that these anomalies may be related to a violation of the faithfulness hypothesis (Runge, 2018a). PCMCI-CMI may fail because the resistivity series ($R1$ to $R6$) are overly deterministically related while accounting for nonlinear dependencies, i.e., without a sufficient stochastic variability. This complementary reason for the instability is further illustrated and detailed in the supplementary materials (SM2.3).

For short datasets, or with unevenly distributed missing values, we consider that building a consensual graph from multiple runs is a good strategy as our results suggested a fast preferential flow in the case of P1 only (Fig. 6.a and b), as expected (Poulain et al., 2018, and Fig. 1). This is, however, computationally expensive. Reducing the number of variables involved in the analysis is another option. It reduces the dimensionality and limits the reduction of the overlapping time-domain due to missing values. Considering a bivariate and nonlinear conditional case (i.e., TE), the low false-positive rate obtained by

Ombadi et al. (2020) is encouraging in that regard. Besides, regarding multivariate approaches, Rinderer et al. (2018) performed the conditioning on the main confounding variables, i.e., effective precipitation, only. This strategy would deserve further consideration, since it makes the study of effective connectivity from multivariate CIMs between two points in the system a systematic and potentially efficient "three-body problem" representation.

We generally consider that multivariate nonlinear CIMs are preferred to assess effective connectivity based on the theoretical background and the results obtained from our synthetic and real experiments. Still, PCMCI-CMI may miss effective connections because of its low recall evidenced by the virtual case (Table 3) and the low number of arrows reported in our consensual graph (Fig. 6). In addition, other assumptions may be violated (Runge, 2018a). In particular, for a real case study, one cannot test the hypothesis of causal sufficiency; that is, all common drivers should be included in the analysis. In short, we

consider causal sufficiency to be challenging to test and conceptualize since hydrological variables can also be represented in a spatially explicit manner in a high-dimensional continuum. CIMs go hand in hand with a dimension reduction task like ours (Delforge et al., 2020b). In the end, there remain conceptual uncertainties and a risk that these CIMs may provide relationships or connections that are spurious and not effective.

### 5.2    Further limitations, recommendations, and perspectives

For our comparative study, we limited ourselves to specific methods and set aside some of their hypotheses to focus on fewer relevant elements for potential CIMs' users in hydrology. More details can be found regarding the underlying frameworks of the newly introduced CIMs, for CCM and/or the chaos theory (Kantz and Schreiber, 2003; Sugihara et al., 2012; Sivakumar, 2017; Huffaker et al., 2018), the PCMCI framework (Runge, 2018a; Goodwell et al., 2020), or both (Runge et al., 2019a). In particular, we assumed but did not discuss stationarity and its various definition depending on the framework. We did not test

the significance of our results using surrogate data test, which stands as a common practice (e.g., Schreiber and Schmitz, 2000; Huffaker et al., 2018). We did not address the impact of noise quantitatively (see Ombadi et al., 2020, for that matter). Also, we hypothesized sustained interactions between variables and did not discuss dynamic intermittency, which could be explicitly visible, such as zeros values in rainfall time-series (Sivakumar, 2017), or hidden and imputed to threshold effects (Blöschl and Zehe, 2005). Arguably, hydrological connectivity is highly dynamic and potentially intermittent with some portion of the

hydrological system getting connected and disconnected depending on its state (Bracken et al., 2013). To explore dynamic intermittency, applying the CIMs on hydrograph segments (e.g., high or low flows; see also Delforge et al., 2020a), or for different seasons (e.g., Ombadi et al., 2020) are two potential options provided that the sample size requirements are met.

     Besides, the selected CIMs operate on the time-domain, limiting us to studying close temporal connections. Yet, hydrological connections and processes can be spread over much longer time-scales. This further questions our tacit assumption of causal

sufficiency. Accordingly, studying methods that operate on the frequency domain or that couple the frequency domain with the time-domain may deserve a particular interest (e.g., Granger, 1969; Molini et al., 2010; Rinderer et al., 2018).

     The PCMCI algorithm is still actively developed. A more recent implementation of PCMCI is now available in the new version of the Tigramite Python package (v4.2), with refined default parameters and improved computational performance. In particular, a new algorithm, PCMCI+, deals with contemporaneous links and strong auto-correlation in series, with the

promises of stronger recall and well-controlled false-positive rate (Runge, 2020). Besides auto-correlation, we found that contemporaneous links are numerous and compromise the recovery of causal direction based on the principle of priority. As contemporaneous links may concern hydrological systems at all spatiotemporal scales, we recommend exploring PCMCI+ for future studies. In particular, while using PCMCI-CMI or any CIM rooted in the information theory, specific attention should be paid to the issue of small sample size (or temporal gaps from missing values). We did not provide any heuristic values linking robustness and the sample size. Such an indicator is contextually linked to the case study and depends on many elements. From the scope of statistical inference with PCMCI, it depends on the Parents' preselection procedure (PC and its parameter $\alpha_{PC}$), the CMI estimator and its parameters, the number of variables, the maximum causal delay $d_{max}$, the nature of the dependencies through their joint probability distributions, or signal/noise characteristics and ratios. From the scope of the hydrological system, it relates to its complexity reflected in the spatiotemporal scale of the hydrological problem, the heterogeneity of its geomorphology (or the patterns of structural connectivity), the nonlinearity of hydrological interactions, their number and spatial organization. Hence, we instead recommend testing the robustness of the results for each particular case study. Potentially, combining any CIM with virtual models mimicking the dynamic properties of the data while knowing the actual causal graph may help select the right CIM, its related estimator, and adequate parametrization. Yet, our toy model (Fig. 2) was not meant for that purpose but to evaluate different CIMs with a parsimonious representation of the confounding problem in hydrology.

A final consideration is more epistemological: should hydrological connectivity (or causality) be studied from a purely empirical and single automated perspective, as with CIMs? We remind that all types of methods can contribute to our causal understanding of environmental systems, e.g., dye tracing tests or spatially detailed inverse resistivity models (Bakalowicz, 2005; Watlet et al., 2018; Poulain et al., 2018). However, the potential sensitivity of CIMs to their assumptions, parameters, or conceptual uncertainties makes them hazardous to use alone, despite their recent noticeable improvements. For this reason, even if PCMCI-CMI appeared to stand out, we recommend comparative studies using several CIMs. Linear CIMs may remain attractive for their intelligibility and computational efficiency. Bivariate methods remain helpful to realize what functional connections multivariate CIMs exclude. We further consider that CIMs complement other methods beyond time-series analysis, e.g., field experiments or physically-based approaches, and they could be combined to narrow the range of possible causal representations of the system under study.

Regarding CIMs only, Klemeš (1982) was particularly critical of letting the data speak for itself. Two avenues are possible and should be kept in mind. As mentioned for CCF in section 5.1.1, we see no issues in applying a bivariate method, with awareness of its limits and appropriate hypothesis testing, in a sequential way, e.g., while removing confounding factors using either a statistical or a physically-based model. That is a supervised causal inference. Another option is to integrate physical constraints in the causal inference algorithm. Here, we adopted the CIM's philosophy of letting the causal graphs be inferred from the data alone. For PCMCI, we have not prescribed any constraint on the conditioning of variables. Yet, physically irrelevant Parents (Eq. 2) may negatively impact the causal graph. Constraining must be framed to prevent us from forcing CIMs on perceptually biased assumptions on the systems' functioning. In particular, this could be done by reintroducing some physical concerns or spatial dimensions into the analysis. Rinderer et al. (2018) already proposed to constrain CIMs

with structural connectivity. However, structural connectivity is unknown, hidden, or too complex to be hypothesized for most karst systems (Bakalowicz, 2005; Hartmann et al., 2014). Still, mass balances, energy potentials, or spatial attributes such as distances or flow path lengths, none currently matter in CIMs. The spatial dimension is initially present in Hume's contiguity principle in time and space (Hume, 1748). To Schrodinger, the spatial continuum is also a causal paradigm in physics (Schrödinger, 1954). Then, we recommend research avenues on reconciling CIMs with space and physics in geosciences.

## 6   Conclusions

The results highlighted that the nonlinear multivariate method, PCMCI coupled with the Conditional Mutual Information test (PCMCI-CMI), shines by its low false-positive rate relative to the other three methods. Hence, statistical dependencies revealed by PCMCI-CMI are more likely to be effective hydrological connections. This advantage is particularly valuable since hydrological systems present highly interdependent time series (or functional connections), favoring a high false-positive rate. This finding confirms our introduced expectation that multivariate nonlinear CIMs are best suited to infer effective connectivity while dealing with nonlinear dependencies and confounding factors resulting from seasonality or meteorological forcing. However, PCMCI-CMI has a low recall, i.e., it misses effective connections, and particular attention should be paid to the robustness of the outcome for small sample sizes or temporal gaps in the data, as evidenced in our real case study. Furthermore, PCMCI-CMI relies on challenging hypotheses to test (Runge, 2018a). Given these uncertainties, PCMCI-CMI, like any other CIM, would always present a risk of spurious results. For this reason, we do not discourage the use of other CIMs as well, for a comparison purpose, with awareness of their limits. Alongside other limitations, recommendations, and perspectives, we remind that framing the causality of hydrological systems is not restricted to the use of the strictly empirical CIMs, but benefits from the vast panel of scientific investigation techniques (e.g., Bakalowicz, 2005; Bracken et al., 2013). In line with Rinderer et al. (2018), hybrid approaches could be developed by reintroducing the physical aspects of the problem to exclude or control the risk of CIMs physically irrelevant outcomes.

*Code and data availability.* CCF and the Student's t-test are computed using the Scipy Python package (Virtanen et al., 2020). The CCM python implementation is available from Delforge et al. (2020a). The official R implementation is available from the CRAN repository: https://CRAN.R-project.org/package=rEDM. PCMCI and independence tests are implemented within the Tigramite (v.4.1 in this case) Python package: https://jakobrunge.github.io/tigramite/. Evapotranspiration data were obtained from the agrometeorological PAMESEB network for the station of Jemelle: https://agromet.be. All other environmental time-series can be obtained from Watlet et al. (2018) and the related repository: https://zenodo.org/record/1158631. Resistivity clustered time-series can be reconstructed following Delforge et al. (2020b) and the example available from the repository: http://dx.doi.org/10.17632/zh5b88vn78.2

.

*Author contributions.* Conceptualization, D. Delforge; methodology, D. Delforge and O. de Viron.; formal analysis, D. Delforge.; investigation, A. Watlet.; data curation, A. Watlet; writing—original draft preparation, D. Delforge.; writing—review and editing, O. de Viron, M. Vanclooster, M. Van Camp, A. Watlet; supervision, M. Vanclooster, M. Van Camp. All authors have read and agreed to the published version of the manuscript.

*Competing interests.* The authors declare that they have no conflict of interest.

*Acknowledgements.* This work is part of a Ph.D. supported by a FRIA grant from the Belgian Fund for Scientific Research (FSR-FNRS). The publication in an open access journal has been supported by the sector of science and technology of UCLouvain. A. Watlet publishes with the permission of the Executive Director, British Geological Survey (UKRI-NERC). We are thankful to M. Van Ruymbeke who designed and constructed drip discharge sensors utilized in this study and to O. Kaufmann who also constructed drip discharge sensors, maintained the underground monitoring system and conceptualized the time-lapse ERT experiment. The authors would like to thank the four anonymous reviewers whose insightful comments significantly improved the quality of the manuscript.

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
