# Peer review of "Detecting hydrological connectivity using causal inference from time-series: synthetic and real karstic study cases"

_Hydrology and Earth System Sciences, 2021_

## Referee Comment (RC2)

**Summary**

This paper by Delforge et al. titled "Detecting hydrological connectivity using causal inference from time-series: synthetic and real karstic study cases" uses four causal inference methods to obtain insights on hydrologic connectivity in a karstic site. The authors also use a synthetic case study to obtain a preliminary understanding on the performance of the methods prior to applying them to the real-world case study.

Overall, the paper objective and the experimental design are well articulated and set clearly. I also think that causal inference is an important topic that should receive more attention from the hydrologic community, and it is of an interest to the audience of HESS. However, I noted some technical issues on the implementation of causal inference methods and the interpretation of results. Also, the discussion section is lacking as the authors didn't compare their findings to recent papers that utilized causal inference in hydrology. Below I summarize these issues and other major comments followed by minor ones. I urge the authors to pay attention to these comments while revising their manuscript. Generally, I recommend that this paper undergoes major revisions before it can be considered for publication.

**Major comments**

1- In the application of multivariate causal inference methods (Partial Correlation and Conditional Mutual Information), the authors didn't illustrate whether the history of variables is used in the conditioning set of variables or not. For instance, if one wants to test for the hypothesis that $Y$ causes $X$ using multivariate methods, then ideally the following conditional test should be implemented: $\mathbb{I}(Y_t, X_t \mid Y_{t-1}:Y_{t-\tau}, X_{t-1}:X_{t-\tau}, Z)$. This conditional test means that one is testing for the statistical relationship between $Y$ and $X$ at time $t$ while conditioning in the history of both variables up to a lag time of $\tau$ as well as the set of variables $Z$ which includes all other confounders. This is a crucial point in the implementation of multivariate methods because it removes the effect of autocorrelation. There are many ways of conditioning in the history of variables; for example, the classical Granger causality (Granger, 1969) conditions in the history of $X$ but not $Y$. Similarly, Transfer Entropy (Schreiber, 2000) does the same, whereas methods such as momentary information transfer (Pompe & Runge, 2011) conditions on both variables. I think that the authors need to mention explicitly what is the conditioning test for both multivariate methods. Please consider adding this information with clear mathematical expressions. Also, the parameter $\tau$ is often one of the most important hyperparameters in causal inference methods, so a discussion on the value of this parameter needs to be included. Please note that $\tau$ in this context is slightly different than $d_{max}$ which the authors use to set the maximum lag time for testing interactions.

2- Related to the previous point, the reason why conditioning in the history of variables is important is because hydrologic timeseries of variables are often highly autocorrelated which leads to spurious causal links, and this testing removes autocorrelation. This context is important in interpreting the results obtained from causal inference methods either in the synthetic or real-world case studies. The authors used first-order difference of the original

time series to remove the effect of seasonality and autocorrelation; however, this is not needed if the implementation of multivariate methods already conditions on the history of variables. Please revise the interpretation of results both in sections 3.2 and 4.2 to highlight this issue of autocorrelation.

3- I found the discussion section to be lacking and it does not report any insightful comparisons to previous work of causal inference in hydrology. For instance, Ombadi et al. 2020 used four causal inference methods on a synthetic and real-world case studies with formal investigation on the impact of sample length, observational and process noise. Some of the methods used in this paper (e.g., CCM and CMI) were also used in that study. It would be important to compare the findings of both studies on the performance of different causal inference methods as this will allow us to build a consensus on the suitability of causal inference methods for hydrologic applications. Although Ombadi et al. 2020 is perhaps the most relevant to this study, there are other studies that used specifically information-theoretic approaches for hydrologic systems characterization such as (Jiang & Kumar, 2019). Please enrich the discussion section by linking the findings of this study to previous work.

4- The record length of the timeseries used in this study is relatively short. This is one of the main challenges that face the application of causal inference methods in hydrology. In my experience, I found that methods based on information theory (e.g., CMI) often needs a long record (~ 2000 – 3000 data points) to provide reasonable performance. The results shown in this study either for the synthetic or real-world case studies are perhaps significantly impacted by the record length yet no discussion was included on the effect of record length. Please enrich the discussion section by highlighting the potential impacts of sample length.

5- Overall, the paper needs to be subjected to proofreading to eliminate grammatical deficiencies and typos as well as to improve the readability. I highlight some of these grammatical errors, unclear sentences and typos in the minor comments below, but these are only few examples and similar corrections should be implemented throughout the paper.

6- Some of the basic information on causal inference methods that was mentioned in section 2 is not accurate and incorrect. For instance, the description of partial correlation in lines 136-138 is not accurate. The authors mention that "partial correlation is like Granger causality". This is quite vague. What the word "like" means specifically here? It is true that partial correlation shares similarities with Granger causality in the sense that both use linear regression to assess interactions while conditioning on potential confounders. However, there are crucial difference too. For instance, Granger causality is technically implemented in a different way than partial correlation with setting both restrictive and unrestrictive regression models, and testing for statistically significant differences using t-test and F-test. These are very crucial differences. Also, on a higher level, the concept of Granger causality is based on what is known as predictive causality and it take into account time precedence. The authors should be careful in introducing the different methods and use precise information. I suggest that the authors revise section 2.1 by writing clear

mathematical expressions for each causal inference method, and also be more precise in their description.

**Minor comments**

1- Lines 315-320 and elsewhere: the instability of CMI here is attributed to missing data. This might be one of the reasons, but I suspect that the main reason is the short record length (see my major comment #4). Also, a possible but unlikely reason is that the instability is the result of changes in the dynamic connectivity. This might be true if the timeseries used in Figure 6 (a, b, c and d) correspond to different hydrologic conditions (wet vs dry). If this latter case is possible, then it is worth of highlight and discussion.

2- Figures 4 & 5: there are several causal links with arrows pointing toward RF (Rainfall)?? Apparently, this is physically incorrect, but I was not able to find any discussion on this in the paper. Are these arrows drawn in the wrong way? Or these are the real results obtained from causal inference methods? If it is the latter case, then this needs to be discussed. In general, this raises a red flag on the accuracy of causal links obtained from the different methods.

3- Lines 228-229: the standard deviation of the noise added to the precipitation signal is unrealistically large!! Even the smallest value used here which is 0.05 of the standard deviation of precipitation is still large. A proportion of 0.001 of the standard deviation of precip is often sufficient to satisfy the condition of causal sufficiency. If process noise is very large, this will impact the results. See Ombadi et al. 2020 for the impact of process noise on the performance of different causal inference methods.

4- Lines 230-231: why only the last year was used? Is it a spin-off period to eliminate the impact of initial conditions or for computational reasons?

5- Lines 252-253: this is perhaps related to the conditioning on the history of variables (see my major comments #1 and #2)

6- Lines 260-262: This is a well-known issue with causal inference methods. You can refer to some studies that pointed to the same issue in evaluating causal inference methods either in hydrology or other fields.

7- Table 4: please replace the abbreviations with the full name (e.g., TP: True positives) or alternatively add this info to the caption of the table so that it can be a standalone component.

8- Figures 4, 5 and 6: I suppose that the numbers in the arrows denote the lag time of interaction in days; however, this was never introduced or mentioned in the captions. Please revise.

9- Lines 27-29: some applications of causal inference in hydrology are missing here. For instance, soil moisture-rainfall feedback (Wang et al., 2018) or differential impact of environmental drivers of evapotranspiration (Ombadi et al., 2020). There are others too if you look in the literature.

10- Lines 35-44: I liked the distinction between structural, functional and effective connectivity. However, from the text, it was not clear what is meant by the effective connectivity and how it differs from the functional one.

11- Line 71: remove "obtained from". Typo.

12- Line 144: the correct name is transfer entropy not "entropy transfer"

13- Line 150: "computationally expensive and quickly require …". The sentence is not logically correct. Please revise.

14- Line 152: grammatical error in "at section 3". It should be "in section 3"

15- Figure 1 caption: when referring to (a) and (b), please remove the parentheses because it is confusing. Only use the parentheses the first time you introduce them.

16- Line 291: replace "As for CCM" with "Similar to CCM". The sentence does not read well currently.

17-  Line 15: this sentence does not read well at all. I understand what you want to convey, but it needs to be rephrased. Something like "…interactions between variables from timeseries only…etc."

18- Lines 98-99: the description of the parameter alpha_pc is not very clear and intuitive to me. Could you elaborate?

**References**

1- Pompe, B., & Runge, J. (2011). Momentary information transfer as a coupling measure of time series. Physical Review E, 83(5), 051122.

2- Granger, C. W. (1969). Investigating causal relations by econometric models and cross-spectral methods. *Econometrica: journal of the Econometric Society*, 424-438.

3- Ombadi, M., Nguyen, P., Sorooshian, S., & Hsu, K. L. (2020). Evaluation of methods for causal discovery in hydrometeorological systems. *Water Resources Research*, 56(7), e2020WR027251.

4- Schreiber, T. (2000). Measuring information transfer. Physical review letters, 85(2), 461. https://doi.org/10.1103/PhysRevLett.85.461

5- Wang, Y., Yang, J., Chen, Y., De Maeyer, P., Li, Z., & Duan, W. (2018). Detecting the causal effect of soil moisture on precipitation using convergent cross mapping. Scientific reports, 8(1), 12171. https://doi.org/10.1038/s41598-018-30669-2

6- Jiang, P., & Kumar, P. (2019). Using Information Flow for Whole System understanding from component dynamics. Water Resources Research, 55(11), 8305-8329.

---

## Author Comment (AC1)

**Response to Anonymous Referee 1**

**1. General response**

Dear reviewer, on behalf of the co-authors and myself, I would like to thank you for your attention to the preprint manuscript and the helpful comments you made. Below are detailed responses to each of your major and minor comments.

Many points you raised seem related to the lack of precision in the methodology section and, for the discussion section, a lack of clear insights and missing references to existing literature. Regarding the methods, although several of your questions are answered in the supplementary materials (SM), we agree that the manuscript should be self-contained. Accordingly, the revised paper is improved by briefly revising the methodological section while keeping the section succinct. Regarding the discussion, based on your remarks and those of the other reviewers, we propose a rewriting of the section with improved references and comparisons to the literature on the following topics:

1. a summary and appreciation of our results;
2. a particular focus on the estimation of Conditional Mutual Information (CMI) concerning missing values, record length, dimensionality, the nature of the dependencies, or noise;
3. and practical recommendations for the uses of causal inference methods and future research perspectives.

Concerning point 2 and the virtual experiment, the other reviewers strongly encourage us to extend the virtual experiment to study the effect of the sample size and/or the number of variables. This was not explicitly your request, although you expressed concern about the instability of the PCMCI-CMI method and its relation to the record length (Major comment 4). Nevertheless, we have decided not to comply with this request for multiple reasons that are reflected in the revised discussion, as any reader may have the same concerns.

These motivations are numerous and quite related to your major comment 4 about the sample size. First, our study remains a comparative study. Such a focus on the PCMCI-CMI method would rather deserve a separate issue. Also, the conclusion of such an extended virtual experiment is reasonably known *a priori*: the results become more robust with increasing sample length or decreasing number of variables (including delayed ones up to $d_{max}$). We see no reason why a non-trivial conclusion, such as recommendations of sample length as a function of the number of variables, would be transposable to a problem with different characteristics, such as different noise levels, model coupling patterns, signal behavior, or representative scales. In addition to the sample size and dimensionality, the CMI also depends on the nature of the CMI dependencies, smooth or not smooth as it could be expected in systems with highly dynamic connectivity, as well as the magnitude and the characteristic of noise. This CMI dependency and noise vary across spatial and time scales. The results also depend on the methods, for instance, kernel-based or nearest neighbors estimators and their hyperparameters.

Our point, with the synthetic studies, was to show the divergence of the methods on the same – simplistic - case study, not as an answer to the question "what should we do", but rather as an exploration of the behavior of the tested methodology in a case where we can give meaningful interpretations of the results. For each problem on which those methods are used, we consider that a good strategy would be to test the issues met and the insights gained by using fit-for-purpose models mimicking the property of signals they want to study.

Thank you again for your contribution to this discussion,

Damien Delforge

**2. Major comments**

2.1 Major Comment 1

> In the application of multivariate causal inference methods (Partial Correlation and Conditional Mutual Information), the authors didn't illustrate whether the history of variables is used in the conditioning set of variables or not. For instance, if one wants to test for the hypothesis that $Y$ causes $X$ using multivariate methods, then ideally the following conditional test should be implemented: $I(Y_t, X_t | Y_{t-1}: Y_{t-\tau}, X_{t-1}: X_{t-\tau}, Z)$. This conditional test means that one is testing for the statistical relationship between $Y$ and $X$ at time $t$ while conditioning in the history of both variables up to a lag time of $\tau$ as well as the set of variables $Z$ which includes all other confounders. This is a crucial point in the implementation of multivariate methods because it removes the effect of autocorrelation. There are many ways of conditioning in the history of variables; for example, the classical Granger causality (Granger, 1969) conditions in the history of $X$ but not $Y$. Similarly, Transfer Entropy (Schreiber, 2000) does the same, whereas methods such as momentary information transfer (Pompe & Runge, 2011) conditions on both variables. I think that the authors need to mention explicitly what is the conditioning test for both multivariate methods. Please consider adding this information with clear mathematical expressions. Also, the parameter $\tau$ is often one of the most important hyperparameters in causal inference methods, so a discussion on the value of this parameter needs to be included. Please note that $\tau$ in this context is slightly different than $dmax$ which the authors use to set the maximum lag time for testing interactions.

The PCMCI algorithm tests for Momentary Conditional Independence (MCI) between variables, which implies conditioning on the Parents of both variables, that are subsets of the historical variables selected using the PC algorithm. This is further explained in the Supplementary materials following the description given in Runge et al. (2019).

Based on your remark, we agreed that the main manuscript could give more details about the algorithm and the tests. As suggested, the revised methodological section now includes the mathematical description for the independence test with a short description in section 2.1. A longer description will still be available in the supplementary materials. Also, we agree that the method section lacks clarity and that the references to supplementary material are not visible enough. Therefore, we propose a few edits in section 2.1.3 to remind that the PCMCI algorithm and the independence tests are further described in the SM. To facilitate a cross-sectional reading of the paper, we also refer to the test equation mentioned above of the MCI in section 2.1, in sections 2.1.3 and 2..1.4 related to ParCorr and MCI. Finally, as you said, the maximum delay is an important parameter that affects the outcome. Some remarks about this parameter are scattered in the preprint article. In particular, in line L190-195, we mention that it should be large enough in virtue of the causal sufficiency but cannot exceed five days due to missing values. In practice, this delay is often set based on the analysis of bivariate dependencies. We propose to stress the importance of the delay and mention these points earlier in the more appropriate section 2.1. Also, we can refer to figure SM.3 and 4 showing the CCF and CCM bivariate dependencies to motivate a maximum delay of 5 days. Both figures show that most of the dependencies do not sustain beyond four days.

2.2 Major Comment 2

> Related to the previous point, the reason why conditioning in the history of variables is important is because hydrologic timeseries of variables are often highly autocorrelated which leads to spurious causal links, and this testing removes autocorrelation. This context is important in interpreting the results obtained from causal inference methods either in the synthetic or realworld case studies. The authors used first-order difference of the original time series to remove the effect of seasonality and autocorrelation; however, this is not needed if the implementation of multivariate methods already conditions on the history of variables. Please revise the interpretation of results both in sections 3.2 and 4.2 to highlight this issue of autocorrelation.

We agree with your point on auto-correlation. Therefore, we will revise sections 3.2 and 4.2 as suggested. Even if the first-order difference is not needed, we consider it worth illustrating it as we believe that potential causal inference method users are unaware of this issue and its implication. This change will also meet the concerns of reviewer #4, who also requested some clarification and explanations about it.

**2.3 Major Comment 3**

I found the discussion section to be lacking and it does not report any insightful comparisons to previous work of causal inference in hydrology. For instance, Ombadi et al. 2020 used four causal inference methods on a synthetic and real-world case studies with formal investigation on the impact of sample length, observational and process noise. Some of the methods used in this paper (e.g., CCM and CMI) were also used in that study. It would be important to compare the findings of both studies on the performance of different causal inference methods as this will allow us to build a consensus on the suitability of causal inference methods for hydrologic applications. Although Ombadi et al. 2020 is perhaps the most relevant to this study, there are other studies that used specifically information-theoretic approaches for hydrologic systems characterization such as (Jiang & Kumar, 2019). Please enrich the discussion section by linking the findings of this study to previous work.

We thank the reviewer for sharing these additional references, particularly the recent work of Ombadi et al., 2020. In addition, we propose to rewrite this section as proposed in our general response to be, as the commentary suggests, more in line with the existing literature and focused on specific issues as the impact of missing data or sample size, or the number of variables (including lagged ones up to $d_{max}$).

At first glance, the study of Ombadi et al. does not challenge our conclusions but complements them. We were particularly interested regarding the results of the bivariate Transfer Entropy (TE) method as we didn't explore it. The asymptotic behavior for True Positive Rate was moderately comparable to the PC method, and the reduction of the False Positive Rate is quite impressive. TE, we believe, could be less demanding in terms of record length as no conditioning is performed for the other variables and could be a viable potential alternative when the sample size is problematic. The behavior of CCM with respect to noise also allowed us to respond to some concerns of reviewer 3. We will make sure to report and discuss these interesting findings properly. The work Jiang and Kumar (2019) is also very relevant as it uses the PCMCI framework implemented in Tigramite to characterize hydrological systems.

**2.4 Major Comment 4**

The record length of the timeseries used in this study is relatively short. This is one of the main challenges that face the application of causal inference methods in hydrology. In my experience, I found that methods based on information theory (e.g., CMI) often needs a long record (~ 2000 – 3000 data points) to provide reasonable performance. The results shown in this study either for the synthetic or real-world case studies are perhaps significantly impacted by the record length yet no discussion was included on the effect of record length. Please enrich the discussion section by highlighting the potential impacts of sample length.

Our beginning discussion mentions the problem of missing values (L316-320). We consider missing values to be problematic in the same sense as the record length problem. Missing values reduce the overlapping time domain over which conditioning can be performed. This is equivalent to a reduction in available samples, triggering issues similar to those of a short record length. Concerning Conditional Mutual Information (CMI), the adequate record length would depend on the dimensionality of the problem (the number of necessary variables to characterize the system) and the complexity of conditional dependencies (e.g., highly non-smooth dependencies, which could be the very case of intermittent or highly dynamic connectivity), which is not scale-invariant in time and space, and may vary geographically from study site to study site.

To the extent that hydrology deals with problems that are empirically complex to characterize beyond an overly simplistic lumped view, we agree that record length should be relatively large, but to what extent, we believe, is highly variable and case-dependent. It also depends on the method chosen to characterize the CMI. In particular, this is the reason why we chose the nearest-neighbor estimator and the shuffle test of Runge (2018) that is better suited than kernel-based approaches for short record length ($< 1000$), based on numerical experiments covering sample sizes from 50 to 2,000 and dimensions up to 10. Yet, despite the use of method recommended for small records, the real study case of the manuscript is concerned by the pitfall of estimating CMI with short record length resulting in non-robust test results, as you also interpret (minor comment #1). We showed that the robustness could be increased by performing an ensemble of tests. Of course, the robustness of a numerical result is one issue; its reliability in terms of connectivity is another.

Overall, the problem of estimating the CMI in case of missing values, short record length, or high dimensionality was the concern of all reviewers. Therefore, as mentioned in our general response, we propose revising the discussion section to discuss this topic properly.

**2.5 Major Comment 5**

> Finally, there are several places with strange phrasing, or where a term is introduced before it is defined, so there is momentary confusion on whether a reference is missing or the sentence is relevant. I am highlighting some of these that I noticed in the minor line-by-line comments below.

We apologize for these writing problems and thank the reviewer for pointing out some of them. We will make sure to correct them and recheck the whole manuscript carefully.

**2.6 Major Comment 6**

> Some of the basic information on causal inference methods that was mentioned in section 2 is not accurate and incorrect. For instance, the description of partial correlation in lines 136-138 is not accurate. The authors mention that "partial correlation is like Granger causality". This is quite vague. What the word "like" means specifically here? It is true that partial correlation shares similarities with Granger causality in the sense that both use linear regression to assess interactions while conditioning on potential confounders. However, there are crucial difference too. For instance, Granger causality is technically implemented in a different way than partial correlation with setting both restrictive and unrestrictive regression models, and testing for statistically significant differences using t-test and Ftest. These are very crucial differences. Also, on a higher level, the concept of Granger causality is based on what is known as predictive causality and it take into account time precedence. The authors should be careful in introducing the different methods and use precise information. I suggest that the authors revise section 2.1 by writing clear mathematical expressions for each causal inference method, and also be more precise in their description.

As mentioned in response to Major Comment 1 and the general response, the revised methodology section aims at being more specific, but not too much longer, while recalling the additional description in the SM. In this case, you will notice that the questions about the differences between the PCMCI-Parcorr and Granger causality are clearly mentioned in the SM1.3. How crucial the differences between the PCMCI-ParCorr and Granger approaches are is a matter for a debate that, in our opinion, does not belong to the paper's scope. We agree that changing the PC routine, or the test to an F-test or a Wald test are alternatives that will probably change the outcomes of causal analysis without altering the general philosophy. Of course, we may understand that if a choice alters the outcome, it is somehow crucial when it comes to causality. By being more specific in the main manuscript, we hope to provide information that may be of interest to the reader in relation to this question they may have or for a simple question of reproducibility. However, we do not have the arguments for preferring one implementation over another and, therefore, we do not wish to elaborate on the different variants of Granger causality.

**3. Minor Comments**

**3.1 Minor Comment 1**

> Lines 315-320 and elsewhere: the instability of CMI here is attributed to missing data. This might be one of the reasons, but I suspect that the main reason is the short record length (see my major comment #4). Also, a possible but unlikely reason is that the instability is the result of changes in the dynamic connectivity. This might be true if the timeseries used in Figure 6 (a, b, c and d) correspond to different hydrologic conditions (wet vs dry). If this latter case is possible, then it is worth of highlight and discussion.

We agree that the record length is the main reason for the instability and is related to the missing value problem (see the general response and major comment 4). The instability to which we first refer is related to the variable results of the test obtained on the same dataset and on the same time domain given by the time distribution of missing values, the record length, and the maximum delay. Since the time domain on which the conditioning is performed does not vary, the same holds for the hydrological conditions. It is, however, confirmed that for the graphs a, b, c, d of Figure 6, the temporal domain for which the variables are conditioned varies since the datasets vary. Hence, the analysis covers more or less hydrological states according to the size of this domain.

Although not stated clearly in the current preprint, we hypothesized that hydrological connections are perennial, i.e., not intermittent, even if sensitive to the hydrological conditions, i.e., nonlinear. We stress this assumption in the revised document. With this in mind, we might expect the hydrologic connections revealed by the MCI to be more robust as the sample size increases. However, this assumption may not be correct if the connectivity is intermittent or highly dynamic. In this case, we believe that the revealed connections are representative of averaged connectivity and may differ if the time domain of the analysis varies for small sample sizes, as you have suggested. It is worth being notified and discussed together with our hypothesis of constant connectivity over time as opposed to intermittent connectivity. We encourage further studies about intermittency in the perspectives at the end of the discussion.

**3.2 Minor Comment 2**

> Figures 4 & 5: there are several causal links with arrows pointing toward RF (Rainfall)?? Apparently, this is physically incorrect, but I was not able to find any discussion on this in the paper. Are these arrows drawn in the wrong way? Or these are the real results obtained from causal inference methods?

If it is the latter case, then this needs to be discussed. In general, this raises a red flag on the accuracy of causal links obtained from the different methods.

Indeed, some links that are obtained from the causal inference methods are physically incorrect. We report some of them, e.g., in L296-297 in the result section: "*We also denote two upward links to R4 and ET. These links seem physically unrealistic and potentially problematic since the effect of P2 is be removed from these variables, which may alter the whole causal graph.*". We further remind the how problematic it is in the discussion L353-356: "*For the multivariate methods, we have chosen to let the causal graphs be formed from the data. We have not prescribed any constraint on the conditioning of variables. This means that variables can be conditioned on potentially aberrant links, negatively impacting the whole causal graph*".

For more clarity, we propose to discuss this in the first part of the discussion about the summary and general appreciation of the results (see general response).

**3.3 Minor Comment 3**

Lines 228-229: the standard deviation of the noise added to the precipitation signal is unrealistically large!! Even the smallest value used here which is 0.05 of the standard deviation of precipitation is still large. A proportion of 0.001 of the standard deviation of precip is often sufficient to satisfy the condition of causal sufficiency. If process noise is very large, this will impact the results. See Ombadi et al. 2020 for the impact of process noise on the performance of different causal inference methods.

We understand that our process noise may be unrealistic because we did not attempt to mimic a realistic environmental rainfall noise. The noisy rainfall series, $P_{eff,A}$ and $P_{eff,B}$ are intermediate variables that are used in the causal analysis. Ultimately, what matters is the resulting noise in the reservoir discharge series $Q_A$ and $Q_b$. In practice, we adjusted the parameters to our liking by graphically interpreting the discharge differences between consecutive runs of the toy model. The noise parameter may seem large because the unit hydrograph and reservoir act as low-pass filters, removing a significant portion of the injected noise.

Following your remark, we wanted to have a better representation of the impact of the noise level parameter $\varepsilon_{lvl}$. The table below shows the 100-runs average correlation $\bar{\rho}$ between two discharge series of 365 days generated with the same unit hydrograph $H = [0.7, 0.2, 0.1]$ and linear storage discharge equation $Q = 0.1S$ per noise level parameter. In the manuscript, this is equivalent to the average correlation between two $Q_A$ obtained with model configuration 1. These values, although roughly evaluated, seem to us to be a reasonable noise panel to explore, especially given the wide range of scales of possible hydrological applications and where we could imagine any kind of dissimilarities.

| $\varepsilon_{lvl}$ | 0.05 | 0.1 | 0.15 | 0.2 | 0.25 | … | 0.7 |
|---|---|---|---|---|---|---|---|
| $\bar{\rho}$ | 0.96 | 0.87 | 0.76 | 0.66 | 0.57 | | 0.24 |

**3.4 Minor Comment 4**

Lines 230-231: why only the last year was used? Is it a spin-off period to eliminate the impact of initial conditions or for computational reasons?

Yes, the early period was considered as a warming-up period. It will be mentioned. Of course, it also impacts computational time.

**3.5 Minor Comment 5**

Lines 252-253: this is perhaps related to the conditioning on the history of variables (see my major comments #1 and #2)

253-253: "*This finding supports our theoretical assertion: the multivariate nonlinear method is the best suited to address effective hydrological connectivity. Furthermore, the method appears to perform better if seasonality is left present in the time-series*".

As we did with the major comments, we agree here as well. We revised to make sure that this is explicit for the reader.

**3.6 Minor Comment 6**

Lines 260-262: This is a well-known issue with causal inference methods. You can refer to some studies that pointed to the same issue in evaluating causal inference methods either in hydrology or other fields.

260-262: *This is particularly contrasting with other methods and provides a valuable piece of information. However, the high precision comes at the cost of a low recall: CMI misses about half of the actual causal links. On the contrary, ParCorr misses none but has a bad precision, i.e., many false positives.*

It is indeed in phase with Runge's reports on the methods. This also could be related to the results of Ombadi et al, 2020. We will look for other references in hydrology.

**3.7 Minor Comment 7**

Table 4: please replace the abbreviations with the full name (e.g., TP: True positives) or alternatively add this info to the caption of the table so that it can be a standalone component.

We will choose one of the two options to improve the readability of the paper.

**3.8 Minor Comment 8**

Figures 4, 5 and 6: I suppose that the numbers in the arrows denote the lag time of interaction in days; however, this was never introduced or mentioned in the captions. Please revise.

Please, note that all captions mention it: "*An undirected line represents contemporaneous dependencies. Delayed dependencies are shown using directed curved arrows. All corresponding delays d are displayed in the middle of its corresponding arrow*"

We will, however, notify the reader that the delay is expressed in days.

**3.9 Minor Comment 9**

Lines 27-29: some applications of causal inference in hydrology are missing here. For instance, soil moisture-rainfall feedback (Wang et al., 2018) or differential impact of environmental drivers of evapotranspiration (Ombadi et al., 2020). There are others too if you look in the literature.

We thank the reviewer for sharing these additional references, as well as those from Jiang and Kumar 2019. We will update our literature review and ensure that we further compare our results with existing studies.

**3.10    Minor Comment 10**

Lines 35-44: I liked the distinction between structural, functional and effective connectivity. However, from the text, it was not clear what is meant by the effective connectivity and how it differs from the functional one.

35-44: *We refer to the terminology of 35 Rinderer et al. (2018), which is inspired by and borrowed from the field of neurological and brain connectivity (Friston, 2011). There are three types of connectivity: (i) structural, (ii) functional, and (iii) effective connectivity. The structural connectivity is derived from the medium and highlights the potential, static, and time-invariant water flow paths from the geological environment's topography, spatial adjacency, or contiguity. The functional one is dynamic and is retrieved from statistical time-dependencies between local hydrological variables. A statistical association may result from confounding factors, e.g., rainfall acting on two disconnected reservoirs or a shared seasonal pattern. Therefore, dependencies do not necessarily imply factual causation, such as process-based water flows. Then, the functional connectivity is a matter of cross-predictability and still reflects potential rather than actual flow paths for water. CIMs with a multivariate framework address confounding factors. They offer the promises of discriminating functional connectivity from the effective one, which reveals actual flow paths and processes within the system. From the structural to the effective connectivity through the functional one, the search for hydrological connections can be seen as a progressive constraint from the potential paths to the actual paths taken by water.*

For more clarity, we propose a reordering of the paragraph and some edits: "[...] *The functional one is dynamic and is retrieved from statistical time-dependencies between local hydrological variables. Functional connectivity is a matter of cross-predictability and reflects dynamic links between the variables. These dynamic links are potential connections subject to confounding factors, i.e., they may or may not be related to a flow process between variables. Effective connectivity precisely refers to actual connections linked through hydrological processes and flows. Since CIMs with a multivariate framework address confounding factors, they offer the promise of discriminating functional connectivity from the effective one. From the structural to the effective connectivity through the functional one, the search for hydrological connections can be seen as a progressive limitation of the possibilities, from the potential paths to the actual paths taken by water.*

**3.11    Minor Comment 11**

11- Line 71: remove "obtained from". Typo.

Corrected. Thank you for pointing the issue.

**3.12    Minor Comment 12**

12- Line 144: the correct name is transfer entropy not "entropy transfer"

Corrected. Thank you for pointing the issue.

**3.13    Minor Comment 13**

13- Line 150: "computationally expensive and quickly require …". The sentence is not logically correct. Please revise.

Corrected. Thank you for pointing the issue.

**3.14 Minor Comment 14**

14- Line 152: grammatical error in "at section 3". It should be "in section 3"

Corrected. Thank you for pointing the issue.

**3.15 Minor Comment 15**

15- Figure 1 caption: when referring to (a) and (b), please remove the parentheses because it is confusing. Only use the parentheses the first time you introduce them.

Corrected. We will avoid parenthesis or parenthesis with a single letter when referring to the subplots. Thank you for pointing the issue.

**3.16 Minor Comment 16**

16- Line 291: replace "As for CCM" with "Similar to CCM". The sentence does not read well currently.

Corrected. Thank you for pointing the issue.

**3.17 Minor Comment 17**

17- Line 15: this sentence does not read well at all. I understand what you want to convey, but it needs to be rephrased. Something like "…interactions between variables from timeseries only…etc."

Corrected. We now use "from timeseries only" as suggested. Thank you for pointing the issue.

**3.18 Minor Comment 18**

18- Lines 98-99: the description of the parameter alpha_pc is not very clear and intuitive to me. Could you elaborate?

98-99: *The PC procedure has a tuning hyperparameter named $\alpha_{PC}$, which controls the number of potential causes. $\alpha_{PC}$ varies from 0 to 1, where 1 is the less restrictive case which implies not pre-selection.*

The alpha_PC is a significance level used to select the parents in the PC stage of the PCMCI algorithm. We agree that "tuning hyperparameter" is too vague when used alone and will use a succinct description stating explicitly that it is a significant level, closer to the one of Runge et al. (2019): "The main free parameter of PCMCI is the significance level $\alpha_{PC}$ in $PC_1$, which should be regarded as a hyperparameter ..."

More details are available in the SM.

**4. Cited references**

Runge, J.: Conditional independence testing based on a nearest-neighbor estimator of conditional mutual information, in: International Conference on Artificial Intelligence and Statistics, International Conference on Artificial Intelligence and Statistics, 938–947, 2018.

Runge, J., Nowack, P., Kretschmer, M., Flaxman, S., and Sejdinovic, D.: Detecting and quantifying causal associations in large nonlinear time series datasets, 5, eaau4996, https://doi.org/10.1126/sciadv.aau4996, 2019.

Jiang, P. and Kumar, P.: Using Information Flow for Whole System Understanding From Component Dynamics, 55, 8305–8329, https://doi.org/10.1029/2019WR025820, 2019.

Ombadi, M., Nguyen, P., Sorooshian, S., and Hsu, K.: Evaluation of Methods for Causal Discovery in Hydrometeorological Systems, 56, e2020WR027251, https://doi.org/10.1029/2020WR027251, 2020.

---

## Author Response (AR1)

**General response to anonymous reviewers**

In this document, we briefly respond to the reviewers' remarks by pointing to the changes made in the revised manuscript. We did not expand our analyses for reasons justified in the specific responses; however, more than two-thirds of the manuscript has been rewritten to consider the reviewers' comments and, accordingly, improve the description of the methods, better discuss the results, and clarify and better convey the limitations, perspectives, and conclusion of this work. As a result:

- The methodology section is more rigorously detailed, with mathematical formulations, improved references to the existing literature, and more transparency about our approaches relative to those found in the literature;
- The real experiment result section (section 4) has been shortened to convey only information about how to read the results and keep the manuscript within a reasonable length. The interpretation is found in the discussion section, which allows reducing redundancy between the two sections;
- The discussion section better comments on our results in dedicated sections (5.1.1 to 5.1.4), focusing, in particular, on their meaning in terms of hydrological connectivity, and with improved reference to the results found in other comparative studies in hydrology. In parallel, we now include a section about the limitations, recommendations, and perspectives (section 5.2) to clarify our message, especially regarding the reviewers' concerns that future readers may share;
- The introduction and conclusion were revised and shortened to be more straightforward.
- The other sections, i.e., the study site and data (section 2.2) and the virtual experiment (section 3), have undergone minor revisions, primarily to address the comments below.

Due to the magnitude of the revisions, we have not justified the changes in the track changes version. Nevertheless, our specific responses below refer to line numbers in the revised manuscript (not the tracked changes) to indicate where the comments have been addressed. The comments are indexed using the following notation, e.g., R1Ma1 for "Reviewer 1 major comment 1", or R2Mi3 for "Reviewer 2 minor comment 3".

**R1Ma1 -Anonymous reviewer 1 - Major comment**

*In the application of multivariate causal inference methods (Partial Correlation and Conditional Mutual Information), the authors didn't illustrate whether the history of variables is used in the conditioning set of variables or not. For instance, if one wants to test for the hypothesis that Y causes X using multivariate methods, then ideally the following conditional test should be implemented: $I(Y_t, X_t \mid Y_{(t-1)}:Y_{(t-\tau)}, X_{(t-1)}:X_{(t-\tau)}, Z)$. This conditional test means that one is testing for the statistical relationship between Y and X at time t while conditioning in the history of both variables up to a lag time of $\tau$ as well as the set of variables Z which includes all other confounders. This is a crucial point in the implementation of multivariate methods because it removes the effect of autocorrelation. There are many ways of conditioning in the history of variables; for example, the classical Granger causality (Granger, 1969) conditions in the history of X but not Y. Similarly, Transfer Entropy (Schreiber, 2000) does the same, whereas methods such as momentary information transfer (Pompe & Runge, 2011) conditions on both variables. I think that the authors need to mention explicitly what is the conditioning test for both multivariate methods. Please consider adding this information with clear mathematical expressions. Also, the parameter $\tau$ is often one of the most important hyperparameters in causal inference methods, so a discussion on the value of this parameter needs to be included. Please note that $\tau$ in this context is slightly different than $dmax$ which the authors use to set the maximum lag time for testing interactions.*

The PCMCI test is now explicitly mentioned (Eq. 2, L145) in our rewriting of the methodological section (section 2.1.3). The revision also aims to introduce GC and TE better. In response to this comment, our description of PCMCI is also more detailed and aims to better describe the method in relation to other existing methods: PC, GC, TE, ... (See L171-186).

**R1Ma2 -Anonymous reviewer 1 - Major comment**

*Related to the previous point, the reason why conditioning in the history of variables is important is because hydrologic timeseries of variables are often highly autocorrelated which leads to spurious causal links, and this testing removes autocorrelation. This context is important in interpreting the results obtained from causal inference methods either in the synthetic or real-world case studies. The authors used first-order difference of the original time series to remove the effect of seasonality and autocorrelation; however, this is not needed if the implementation of multivariate methods already conditions on the history of variables. Please revise the interpretation of results both in sections 3.2 and 4.2 to highlight this issue of autocorrelation.*

We revised section 3.1 to add the purpose of differencing (L268): "The primary purpose of the differenced data is to simply illustrate the effect and value of removing past dependencies (auto-correlation, seasonality) on the CIMs results. "

We revised section 3.2 to inform that differencing is indeed not necessary for multivariate methods (L280): "... differencing time-series is theoretically useless as PCMCI already conditions on the past of variables (Eq. 2) and is capable of dealing with auto-correlation and seasonality"

We revised section 4.2 (L310): "we chose to report causal graphs for the raw (not differenced) data since differencing is unnecessary and reduced the precision of CMI in the virtual experiment (Table 3)"

**R1Ma3 -Anonymous reviewer 1 - Major comment**

*I found the discussion section to be lacking and it does not report any insightful comparisons to previous work of causal inference in hydrology. For instance, Ombadi et al. 2020 used four causal inference methods on a synthetic and real-world case studies with formal investigation on the impact of sample length, observational and process noise. Some of the methods used in this paper (e.g., CCM*

*and CMI) were also used in that study. It would be important to compare the findings of both studies on the performance of different causal inference methods as this will allow us to build a consensus on the suitability of causal inference methods for hydrologic applications. Although Ombadi et al. 2020 is perhaps the most relevant to this study, there are other studies that used specifically information-theoretic approaches for hydrologic systems characterization such as (Jiang & Kumar, 2019). Please enrich the discussion section by linking the findings of this study to previous work.*

The discussion section has been completely revised to better discuss our results and relate them to the existing literature. We mostly compare them to the study of Ombadi et al. 2020 and Rinderer et al. 2018, because these are the only comparative study in hydrology, with some reports on the performance, e.g., false-positive rates, that we found.

**R1Ma4 -Anonymous reviewer 1 - Major comment**

*The record length of the timeseries used in this study is relatively short. This is one of the main challenges that face the application of causal inference methods in hydrology. In my experience, I found that methods based on information theory (e.g., CMI) often needs a long record (~ 2000 – 3000 data points) to provide reasonable performance. The results shown in this study either for the synthetic or real-world case studies are perhaps significantly impacted by the record length yet no discussion was included on the effect of record length. Please enrich the discussion section by highlighting the potential impacts of sample length.*

In our revised section 2.1.3, we discuss sample size requirements (L155-161). In particular: " Based on numerical experiments covering sample sizes from 50 to 2,000 and dimensions up to 10, Runge (2018b) recommends using nearest-neighbors estimators of CMI (Frenzel and Pompe, 2007; Vejmelka and Paluš, 2008) for short sample size (< 1000)". We considered our synthetic results to be stable and this is reflected in Figure 3 and Table 3. The new revised section 5.1.4 elaborates on the problem of small sample size and missing values (L412-420).

**R1Ma5 -Anonymous reviewer 1 - Major comment**

*Finally, there are several places with strange phrasing, or where a term is introduced before it is defined, so there is momentary confusion on whether a reference is missing or the sentence is relevant. I am highlighting some of these that I noticed in the minor line-by-line comments below.*

The manuscript has undergone several revisions for grammar and spelling.

**R1Ma6 -Anonymous reviewer 1 - Major comment**

*Some of the basic information on causal inference methods that was mentioned in section 2 is not accurate and incorrect. For instance, the description of partial correlation in lines 136-138 is not accurate. The authors mention that "partial correlation is like Granger causality". This is quite vague. What the word "like" means specifically here? It is true that partial correlation shares similarities with Granger causality in the sense that both use linear regression to assess interactions while conditioning on potential confounders. However, there are crucial difference too. For instance, Granger causality is technically implemented in a different way than partial correlation with setting both restrictive and unrestrictive regression models, and testing for statistically significant differences using t-test and Ftest. These are very crucial differences. Also, on a higher level, the concept of Granger causality is based on what is known as predictive causality and it take into account time precedence. The authors should be careful in introducing the different methods and use precise information. I suggest that the authors revise section 2.1 by writing clear mathematical expressions for each causal inference method, and also be more precise in their description.*

The entire methodological section has been revised to be more precised. Differences between PCMCI and other methods such as GC are clearly exposed in the new section 2.1.3 (L171-186).

**R1Mi1 -Anonymous reviewer 1 - Minor comment**

> *Lines 315-320 and elsewhere: the instability of CMI here is attributed to missing data. This might be one of the reasons, but I suspect that the main reason is the short record length (see my major comment #4). Also, a possible but unlikely reason is that the instability is the result of changes in the dynamic connectivity. This might be true if the timeseries used in Figure 6 (a, b, c and d) correspond to different hydrologic conditions (wet vs dry). If this latter case is possible, then it is worth of highlight and discussion.*

The new revised section 5.1.4 elaborates on the problem of data length and missing values (L412-420). For clarity, we do not mix the stability issue with the dynamic intermittency problem. Dynamic intermittency is discussed in the new section 5.2 on limitations and perspectives (L451-456), e.g.: "... To explore dynamic intermittency, applying the CIMs on a segmentation of the hydrograph (e.g., high or low flows), or for different seasons (Ombadi et al., 2020) are two potential options provided that the sample size requirements are met ".

**R1Mi2 -Anonymous reviewer 1 - Minor comment**

> *Figures 4 & 5: there are several causal links with arrows pointing toward RF (Rainfall)?? Apparently, this is physically incorrect, but I was not able to find any discussion on this in the paper. Are these arrows drawn in the wrong way? Or these are the real results obtained from causal inference methods? If it is the latter case, then this needs to be discussed. In general, this raises a red flag on the accuracy of causal links obtained from the different methods.*

These are not drawn in the wrong way. We now discuss some of these physically irrelevant links. For instance, in section 5.1.1 (CCF, L344-345): "Some links are more intriguing and are hardly interpreted as a causal relationship, e.g., R5 causing RF." In section 5.1.2 (CCM), we discuss the reported relation between P2 and P1 (L371-373). In section 5.1.3 (ParCorr, L395), we mention that irrelevant links are alarming, even more, because of their negative feedback on the entire causal graph. In the perspective section (5.2, L481-504), we question whether or not one could manually constraint multivariate CIMs to prevent physically irrelevant links.

**R1Mi3 -Anonymous reviewer 1 - Minor comment**

> *Lines 228-229: the standard deviation of the noise added to the precipitation signal is unrealistically large!! Even the smallest value used here which is 0.05 of the standard deviation of precipitation is still large. A proportion of 0.001 of the standard deviation of precip is often sufficient to satisfy the condition of causal sufficiency. If process noise is very large, this will impact the results. See Ombadi et al. 2020 for the impact of process noise on the performance of different causal inference methods.*

As mentioned in our previous response, we do not consider that our noise level is unrealistic given that our model is a toy model and that real case may cover wide range of noise for various spatiotemporal scales. Still, we added the following sentence to provide a better representation of the noise level (L260): " With such noise levels, we estimated that the correlation between two generated Q_A with model configuration 1 (Table 2) would vary on average between 0.96 (eps=0.05) and 0.24 (eps=0.7)". We now refer to Ombadi et al. 2020 to discuss the effect of noise for CCM in section 5.1.2 (L376), and the question: "could CCM be applied on noisy time-series". We further refer to Ombadi et al. 2020 while reminding that we did not discuss quantitatively the impact of noise in our study in section 5.2 (L451).

**R1Mi4 -Anonymous reviewer 1 - Minor comment**

*Lines 230-231: why only the last year was used? Is it a spin-off period to eliminate the impact of initial conditions or for computational reasons?*

We now mention that it is a warming up period (L264).

**R1Mi5 -Anonymous reviewer 1 - Minor comment**

*Lines 252-253: this is perhaps related to the conditioning on the history of variables (see my major comments #1 and #2)*

See R1Ma2

**R1Mi6 -Anonymous reviewer 1 - Minor comment**

*Lines 260-262: This is a well-known issue with causal inference methods. You can refer to some studies that pointed to the same issue in evaluating causal inference methods either in hydrology or other fields.*

In L295, we now refer to Runge, J.: Discovering contemporaneous and lagged causal relations in autocorrelated nonlinear time series datasets, in: Proceedings of the 36th Conference on Uncertainty in Artificial Intelligence (UAI), Conference on Uncertainty in Artificial Intelligence, 1388–1397, 2020.

The problem of low recall is covered, and a potential solution is proposed in the perspective section (5.2), while referring to the same paper (L465).

**R1Mi7 -Anonymous reviewer 1 - Minor comment**

*Table 4: please replace the abbreviations with the full name (e.g., TP: True positives) or alternatively add this info to the caption of the table so that it can be a standalone component.*

Table 4 is now Table 3 (Table 1 was removed in the process of shortening the paper) and contains the full names as suggested.

**R1Mi8 -Anonymous reviewer 1 - Minor comment**

*Figures 4, 5 and 6: I suppose that the numbers in the arrows denote the lag time of interaction in days; however, this was never introduced or mentioned in the captions. Please revise.*

The delay in days is mentioned in the caption and in the main text (L305). The result section has been improved to better convey how the results should be read.

**R1Mi9 -Anonymous reviewer 1 - Minor comment**

*Lines 27-29: some applications of causal inference in hydrology are missing here. For instance, soil moisture-rainfall feedback (Wang et al., 2018) or differential impact of environmental drivers of evapotranspiration (Ombadi et al., 2020). There are others too if you look in the literature.*

We added two references in the introduction: Ombadi et al., 2020, and Wang et al. 2018. We added two more references in the method description: Jiang and Kumar, 2019 (for PCMCI), and Medina et al., 2019 (for CCM).

**R1Mi10 -Anonymous reviewer 1 - Minor comment**

*Lines 35-44: I liked the distinction between structural, functional and effective connectivity. However, from the text, it was not clear what is meant by the effective connectivity and how it differs from the functional one.*

For more clarity, we propose a reordering of the previous paragraph and some edits. See the new definitions in L41-48.

**R1Mi11 -Anonymous reviewer 1 - Minor comment**

*11- Line 71: remove "obtained from". Typo.*

Corrected.

**R1Mi12 -Anonymous reviewer 1 - Minor comment**

*12- Line 144: the correct name is transfer entropy not "entropy transfer"*

We now refer to transfer entropy or TE every time.

**R1Mi13 -Anonymous reviewer 1 - Minor comment**

*13- Line 150: "computationally expensive and quickly require ...". The sentence is not logically correct. Please revise.*

Not applicable anymore as the methodological section has been entirely revised to be more precise.

**R1Mi14 -Anonymous reviewer 1 - Minor comment**

*Line 152: grammatical error in "at section 3". It should be "in section 3"*

Not applicable anymore as the methodological section has been entirely revised to be more precise.

**R1Mi15 -Anonymous reviewer 1 - Minor comment**

*15- Figure 1 caption: when referring to (a) and (b), please remove the parentheses because it is confusing. Only use the parentheses the first time you introduce them.*

Corrected.

**R1Mi16 -Anonymous reviewer 1 - Minor comment**

*16- Line 291: replace "As for CCM" with "Similar to CCM". The sentence does not read well currently.*

Corrected.

**R1Mi17 -Anonymous reviewer 1 - Minor comment**

*17- Line 15: this sentence does not read well at all. I understand what you want to convey, but it needs to be rephrased. Something like "...interactions between variables from timeseries only...etc."*

In the end, we replaced by "… between variables from data only", because we cannot refer to time-series while referring to the work of Spirtes et al. and Pearl. See L16.

**R1Mi18 -Anonymous reviewer 1 - Minor comment**

*18- Lines 98-99: the description of the parameter alpha_pc is not very clear and intuitive to me. Could you elaborate?*

We reformulated in L142-144: "The resulting size of Parents' set [note: from the PC selection step] are controlled by alpha_PC, a liberal parameter varying between 0 and 1, with the latter being the less restrictive case that includes all possible variables."

**R2Ma1 -Anonymous reviewer 2 - Major comment**

> *The lack of statistically significant links in the CMI method made more sense to me when I noticed the sparse dataset (e.g. fewer than 100 data points) that is available for the karst system analysis. In general, I think this method would not be expected to produce robust results given this amount of data, due to the high dimensional pdfs involved for information theory measures. With this, I think the interpretation should not be that CMI performed "worse" in some way, but that it has higher need for data length as a much higher dimensional approach, especially relative to the bivariate methods. Additionally, it seems like the bivariate methods actually do utilize the full amount of data available for any two variables, which means that the comparison is even less "fair" – it might be better to only consider the time window in which there is data available for all (or some, like the P1, P2, and P3 cases in Figures 5 and 6) variables. I think this aspect should be made more clear at the least, and possibly deserves a change in the time windows used for the comparison.*

For reasons exposed in our previous answer, we did not consider that applying CCF and CCM on the full amount of data was unfair. Simply put, not using the full data set for bivariate methods would have been equally unfair, in the sense that it deprives them of an advantage of their own, and from the point of view of the principle of induction, which suggests that a generalization must take into account the full data set available to construct and/or test it.

Regarding PCMCI-CMI, the revised manuscript better covers the sample size issue. The improved methodology better refers to the testing of Runge (2018) (see the new 2.1.3, L157). Our new discussion is much more equivocal on this point (section 5.1.4, L412-420). We did not mean to imply that the PCMCI-CMI is worse in some sense. A revision of the abstract and conclusion presents a more optimistic and correct message: we recommend PCMCI-CMI while notifying that attention should be paid to the sample size issue.

**R2Ma2 -Anonymous reviewer 2 - Major comment**

> *I also have a question about the statistical significance. For example, in some information theory based studies, we use a shuffled surrogates method for statistical significance of a given link, which would differ from a p-value in a correlation analysis. If the method for identifying a statistically significant link varies at all between methods, this also needs to be apparent.*

For CCF (section 2.1.1, L75)), CCM (2.1.2, L99), we mention that a Student t-test is used. For PCMCI-ParCorr (2.1.3), the linear model and the Student t-test is clearly explained (L147-149). For PCMCI-CMI (2.1.3), we also explain that no analytical test is available for our nearest-neighbor estimator of CMI (Runge, 2018).

Later on, in the CCF (5.1.1, L347) or CCM (5.1.2, L368), we mention that surrogate data test (referring to Schreiber, 2000) could be used to better control the number of significant dependencies.

**R2Ma3 -Anonymous reviewer 2 - Major comment**

> *For the synthetic case, I see you used a lot longer dataset than you have available for the real karst case study. This could lead to the better performance for the higher dimensional or multivariate methods – it might be useful to test the synthetic case for a much smaller dataset and observe or confirm whether these methods start to lose their detection of links. This could better show that the CMI/ParCorr types of methods do have better performance, but only when given a lot of data. I think the differences between the methods might make them inherently difficult to compare, but these are some things that could improve the attempt.*

We did not revise the manuscript following the suggestion because, as explained in our previous answer, we believe that such heuristic values could be misleading, and that was not the purpose of our toy model.

Still, we have done the following edits: in section 2.1.2 (CCM, ~L125-129), we mention that, indeed, sample length is an issue for all CIM, to different extent and refer to (Runge et al., 2019, and Ombadi et. al., 2019).

As deemed relevant for future readers, we also reformulated our answer in a paragraph (L468-480), specifying that the question is complex, study-case related, and that our toy model was not meant for that purpose.

**R2Ma4 -Anonymous reviewer 2 - Major comment**

*Finally, there are several places with strange phrasing, or where a term is introduced before it is defined, so there is momentary confusion on whether a reference is missing or the sentence is relevant. I am highlighting some of these that I noticed in the minor line-by-line comments below.*

The manuscript has undergone several revisions for grammar and spelling.

**R2Mi1 -Anonymous reviewer 2 - Minor comment**

*Line 7: "appears unstable" relates to my comment on data length...I think the instability is at least partially due to a very small dataset. Either way, it is not very clear what this term means within the abstract.*

This sentence has been rewritten (L7-9): " However, for the real study case, the multivariate nonlinear method was unstable because of the uneven distribution of missing values affecting the final sample size for the multivariate analyses, forcing us to cope with the results' robustness".

**R2Mi2 -Anonymous reviewer 2 - Minor comment**

*Line 15: "between variables from variables" did not make sense to me*

Rephrase into "between variables from data only". See also R1Mi17

**R2Mi3 -Anonymous reviewer 2 - Minor comment**

*Line 28: cross-scale*

**R2Mi4 -Anonymous reviewer 2 - Minor comment**

*Line 35-45: This paragraph seemed scattered, and I did not come out of it with a clear understanding of "effective connectivity" in particular. Suggest to revise*

Edited. See R1Mi10

**R2Mi5 -Anonymous reviewer 2 - Minor comment**

*Line 41: "process-based water flows" – I'm not sure if there are non-process-based flows of water?*

Modified. See R1Mi10

**R2Mi6 -Anonymous reviewer 2 - Minor comment**

*Line 45: "progressive constraint" was not clear to me*

Removed

**R2Mi7 -Anonymous reviewer 2 - Minor comment**

*Line 50: "heterogeneity" instead of "hiddenness"?*

This paragraph was substantially modified. We now refer to heterogeneity and hiddenness as follows (L53-54): "Assessing structural connectivity in karst systems is a challenging task because of their hidden and heterogeneous structure (Bakalowicz, 2005)"

**R2Mi8 -Anonymous reviewer 2 - Minor comment**

*Line 60: I'm not sure about the sentence "nonlinearity is imputed to nonlinear hydrological processes", seems redundant*

Edited. We now refer to the nonlinearity of karst systems as follow (L55-57): "Besides, karst systems are known for their nonlinear behavior, which could be imputed to nonlinear hydrological processes, e.g., taking the form of power laws, or threshold effects triggering flows (Bakalowicz, 2005; Blöschl and Zehe, 2005)"

**R2Mi9 -Anonymous reviewer 2 - Minor comment**

*Line 65: would be good to re-define CCF here*

For more clarity, all acronyms are defined together in the second paragraph of the introduction (L33-38).

**R2Mi10 -Anonymous reviewer 2 - Minor comment**

*Line 74: What do you mean by "to appreciate the results" – to compare with the results, or validate them?*

Replaced by: " We expect CIMs to reveal this specific connection"(L68).

**R2Mi11 -Anonymous reviewer 2 - Minor comment**

*Line 85: "Being multivariate" – I'm not sure that a multivariate approach inherently deals with confounding effects. For example, a multiple linear regression is multivariate, but does not do any type of conditioning on confounding variables…*

The methodology section has been rewritten to be more precise.

See our previous answer: *We are not sure to understand your point. What we mean here is that, as several causes can be entered in the test, it can distinguish - if the problem is well posed and fulfills the condition of causal sufficiency - between direct and indirect causation. A multilinear regression, such as Granger causality, can do the same. In that sense, we consider that it can cope with confounding variables*

**R2Mi12 -Anonymous reviewer 2 - Minor comment**

*Line 93: PC and MCI are brought up, and then defined later – would be better to re-arrange such that we are not wondering what they are.*

See R2Mi9

**R2Mi13 -Anonymous reviewer 2 - Minor comment**

*Line 99: "not preselection" to "no preselection"?*

No longer applicable since PCMCI description has been entirely revised.

**R2Mi14 -Anonymous reviewer 2 - Minor comment**

> *Line 103: reference for causal sufficiency? In general, this is a good point for any analysis, you particularly reference it for CMI, could state that this hypothesis really underlies all your methods…*

Indeed, causal sufficiency is related to the principle of common cause. We mention it in the new PCMCI section 2.1.3 (L164): "PCMCI is based on a strict framework of assumptions: faithfulness, causal sufficiency, the absence of contemporaneous dependencies, the Causal Markov Condition, stationarity, [...] Causal sufficiency implies that monitored variables include all common causes, following the principle (Reichenbach, 1956).

**R2Mi15 -Anonymous reviewer 2 - Minor comment**

> *Section 2.2.1: I felt like you did some discussion previously that was particular to each method, but then you have these sections for each method separately. I would move some of the above material at the beginning of 2.1 into these sections directly, and save the "causal sufficiency" aspect at the beginning as it applies to any method.*

The revised introduction, methodological, and discussion sections were entirely reworked to be more transparent, precise, and better separated. In the end, we did not introduce causal sufficiency in the introduction as it is not clearly discussed in some causal inference framework as CCM. CCM rather elaborates on the concept of synchrony (now explained in the CCM section 2.1.2, ~L105). We rather refer to the principle of common cause in the introduction, which is more general, and less related to the terminology of the PCMCI framework. See also  R2Mi14.

**R2Mi16 -Anonymous reviewer 2 - Minor comment**

> *Line 121: "overall good performance of this value" is vague, do you mean for the synthetic study, or the real study, or in general?*

For more clarity, we added (L91): "due to the overall good performance of this value during our preliminary testing"

**R2Mi17 -Anonymous reviewer 2 - Minor comment**

> *Line 136: I don't think you have introduced Granger Causality*

The revised methodological section 2.1.3 on PCMCI now correctly introduce GC (L135 and L174). See also R1Ma6.

**R2Mi18 -Anonymous reviewer 2 - Minor comment**

> *Line 153: Is two weeks of computation for a single processor? I figure it would take different amounts of time depending on whether you used a laptop or a server, etc, so could make this more clear.*

In the revision (section 2.1.3, L161), we rather refer to Runge (2018)'s testing while mentioning that the test is computationally expensive. The computational time we experienced is mentioned in the supplementary materials SM2.3, with the associated laptop hardware.

**R2Mi19 -Anonymous reviewer 2 - Minor comment**

> *Line 176: It seems like for 2014-2017 time-series, there would be more than 465 time steps, implying the presence of gaps. This also comes into play in terms of your total data length for the multivariate methods. Basically, the counts in Table 2 make it seem like there is more data available than there actually is, when you start comparing multiple datasets (with 48 data points being the total overlap).*

We hope that the issue of missing values is more cleary exposed in the revised version of the manuscript, with a better description of the conditioning mechanisms in section 2.1.3 (L139 and 156-159), the temporal gaps in the data (section 2.2, L220-231), and the dedicated discussion in section 5.1.4 (L412-434).

**R2Mi20 -Anonymous reviewer 2 - Minor comment**

*Line 204: "problematic case of the common cause" – after this, you define what this means, but as it is, the phrase seems a little mysterious, like a Sherlock Holmes story.*

This has been modified (L236): "a simple hydrological reservoir model is inspired by the common cause problem (Fig. 2)"

**R2Mi21 -Anonymous reviewer 2 - Minor comment**

*Line 210: haven't defined Qb' yet*

This has been rephrased (L241): "For comparison, we consider a case where $Q\_A$ is effectively connected to another series $Q\_B'$ ... "

**R2Mi22 -Anonymous reviewer 2 - Minor comment**

*Line 230: This is a "synthetic study" but the years make it seem like you are using actual data from your real study?*

We added "computed from real data (section 2.2., ~L245)" in the caption of Fig 2.

**R2Mi23 -Anonymous reviewer 2 - Minor comment**

*Line 234: What is a differenced dataset? This comes up a few times and I'm not completely sure what it is...whether it is the increment or something done in the modeling process.*

We added ", i.e., $Y\_t - Y\_{t-1}$," to be explicit (L267).

**R2Mi24 -Anonymous reviewer 2 - Minor comment**

*Line 270: "If causality is hard to infer…" is not a great sentence, excusing a complicated figure and telling us it actually makes sense. You could just remove this.*

Such formulation does not appear anymore in the revised discussion of CFF (secton 5.1.1)

**R2Mi25 -Anonymous reviewer 2 - Minor comment**

*Line 297: "is be removed"*

Such formulation does not appear anymore in the revised discussion of ParCorr (section 5.1.3)

**R2Mi26 -Anonymous reviewer 2 - Minor comment**

*Line 338: "evanescent singularity" is unclear*

The sentence is removed from our revised discussion.

**R3Ma1 -Anonymous reviewer 3 - Major comment**

*Comparison between CCM and CMI-based PCMCI. The current comparison based on noisy data is unfair to CCM, because CCM is more suitable for deterministic dynamics and does not work well in a*

*stochastic system. Also, the authors used different levels of noises in the synthetic study. Still, only the averaged results are reported in Figure 3, and the noise impact on the performances of the four methods remains unknown. Therefore, I suggest, at least in the synthetic case study, performing the comparison based on a noise-free/deterministic system and a thorough evaluation of the noise impact.*

As mentioned in our response, we consider the idea of a CCM limited to deterministic dynamics to be advanced in the literature but, in our opinion, too conservative. Our opinion is more apparent in the new version and related to the littérature: see the last two paragraphs of section 2.1.2 (L100-129) and CCM discussion (5.1.2). In particular, we refer to Ombadi et al. (2020) for the evaluation of the effect of noise.

**R3Ma2 -Anonymous reviewer 3 - Major comment**

*Limited data points for computing CMI. In the real case study, the inferred causality from CMI-based PCMCI is much less trustable, given only 465 datapoints of 7 variables (what is the maximum allowed number of conditioned variables set in PCMCI by the way?). In fact, it is somehow expected that the CMI-based PCMCI does not work well using this limited dataset (even for a three-dimensional CMI estimation, several hundred data points might not be sufficient). Although the authors acknowledged this limitation, I strongly recommend a corresponding synthetic study to evaluate the impact of dataset size and the number of variables in CMI-based PCMCI, which is very critical to guide the current and future causality analysis in earth science inferred by the PCMCI algorithm.*

This comment relates to the concerns of the other reviewer as well. See R1Ma4, R2Ma1, and, in particular, R2Ma3.

**R3Mi1 -Anonymous reviewer 3 - Minor comment**

*Lines 67 and 68: Please spell out ParCorr and CMI.*

For more clarity, all acronyms are defined together in the second paragraph of the introduction. See also R2Mi9

**R3Mi2 -Anonymous reviewer 3 - Minor comment**

*Line 144: "entropy transfer" –> "transfer entropy"*

Corrected. See also R1Mi12

**R3Mi3 -Anonymous reviewer 3 - Minor comment**

*Lines 147 and 148: "The nearest-neighbor estimator is recommended for time-series below 1000 samples"... under what dimensionality?*

This is now mentioned in section 2.1.3 (L158) and discussed in section 5.1.4 (L415-420). See R1Ma4.

**R3Mi4 -Anonymous reviewer 3 - Minor comment**

*Line 238: Qb -> QB*

Corrected. B in now capitalized

**R3Mi4 -Anonymous reviewer 3 - Minor comment**

*Line 356: "constraint causal inference" –> "constrain causal inference"*

No longer applicable and corrected in the revised discussion. Constrain/constraint is in its correct form everywhere in the manuscript.

**R4Ma1 -Anonymous reviewer 4 - Major comment**

> *I was confused about the difference between the original dataset and the differenced dataset when comparing the outcomes of the methods. I am not sure what was differenced to produce such different results within the method. Further clarification on this would be great.*

Same as R2Mi23. We added ",  i.e., Y_t - Y_{t-1}," to be explicit (L267).

**R4Ma2 -Anonymous reviewer 4 - Major comment**

> *When you find contemporaneous links, could this be due to shorter term processes that could be resolved with a shorter time step? So an instant link means that the data are already synchronized and presumably there exists a measurable scale that could capture the actual lag of that information flow. Is that possible for the karstic data?*

As mentioned in our previous answer, we consider contemporaneous links to be an issue at all scales. In the new manuscript section 5.2, we mention the following perspective (L464-468): " In particular, a new algorithm, PCMCI+, deals with contemporaneous links and strong auto-correlation in series, with the promises of stronger recall and well-controlled false-positive rate (Runge, 2020). Besides auto-correlation, we found that contemporaneous links are numerous and compromise the recovery of causal direction based on the principle of priority. As contemporaneous links may concern hydrological systems at all spatiotemporal scales, we recommend exploring PCMCI+ for future studies..".

**R4Ma3 -Anonymous reviewer 4 - Major comment**

> *Discussion section: The discussion section could be improved by highlighting the results in terms of the connectivities described in the introduction, especially for the real case study. Also, greater connections between these results and previous studies on CIMs would add better context to the contributions of this study.*

Also in response to R1Ma3, the discussion section has been entirely revised to be more insightful and related to other comparative studies in hydrology (i.e., Rinderer et al, 2018; Ombadi et al., 2020). For each method, we specifically discuss (section 5.1.1 to 5.1.4) the results in terms of connectivity.

**R4Mi1 -Anonymous reviewer 4 - Minor comment**

> *L66: You state the abbreviation of the method before stating the actual name. Please correct*

For more clarity, all acronyms are defined together in the second paragraph of the introduction (L33-38). See also R2Mi9

**R4Mi2 -Anonymous reviewer 4 - Minor comment**

> *L70: The phrase "obtained from" is repeated twice.*

Corrected. Same as R1Mi11.

**R4Mi3 -Anonymous reviewer 4 - Minor comment**

> *L99: Change "not" to "no"*

The methodological section has been entirely revised, and this mistake is no longer present.

**R4Mi4 -Anonymous reviewer 4 - Minor comment**

> *Figure 1 caption: It is unclear what the red areas are showing in the figure based on the description. Is it the overlapping time-spans for all data? Or just the portion that can be analyzed using a 5-day lag? Please clarify.*

The last paragraph of section 2.2 (L220-231) has undergone minor edits to clarify that the red areas are indeed overlapping time-span given a lag of 5 days.

The reduction of the final time-domain is also better explained in the methodological section of PCMCI (2.1.3, L139) and its implications in terms of stability for PCMCI-CMI in section (5.1.4, L412-424).

**R4Mi5 -Anonymous reviewer 4 - Minor comment**

> *L210: What is QB'?*

QB' is now properly introduced. See R2Mi21.

**R4Mi6 -Anonymous reviewer 4 - Minor comment**

> *L218: Move the phrase "with R either A or B" earlier, when you first introduce HR.*

Corrected as suggested.

**R4Mi7 -Anonymous reviewer 4 - Minor comment**

> *L225, that paragraph: What is the length of the dataset? How did you set your length to ensure sufficient data for applying the CIMs?*

The length of the dataset is mentioned in L263 (one year).

See our previous answer: "*The length is 365 days (L230). We did not ensure this but did not notice any particular unstable behavior, the performance was satisfactory and fully detailed in Table 2 [erratum: mistake, in the revised version Table 3]. Possibly, we would have obtained better results with longer dataset. In the revised perspective and recommendation, we recommend using a virtual case mimicking the signal properties to answer this type of case specific questions.*"

The above-mentioned recommendation is found in the perspective section (5.2), in the end of the 3rd paragraph (~L473).

**R4Mi8 -Anonymous reviewer 4 - Minor comment**

> *L233: Formatting issue for variable HAB.*

Corrected.

**R4Mi9 -Anonymous reviewer 4 - Minor comment**

> *L238, that paragraph: It would help to reference specific parts of Figure 3 in the paragraph*

We now reference the panels in the main text: "...obtained with the four CIMs (a to d)" (L270 and below).

**R4Mi10 -Anonymous reviewer 4 - Minor comment**

> *L271: Are the patterns shown in the figure or just stated here? It is difficult to know which relationships you are showing. I am also confused by what you mean by the time dependencies flipping as you can't have a negative delay? Please clarify.*

To avoid redundancies and clarify the paper, we restructured such that the results aim at conveying how to read the outcomes presented in the Figures. Any interpretation of the result is found in the specific discussion section. For CCF and the present comment's concern, we meant that the sign of the correlation is flipping. We reformulated in section 5.1.1, L340:

"[...] Often, positive relationships, e.g., R5 -> P2 with d={2, 3}, follow negative ones that are more interpretable as a transfer (e.g., R5 -> P2 with d= {0, 1}). This pattern is rather a phasing artifact captured by CCF interpretable as: "after the rain, the good weather", and vice-versa. "

**R4Mi11 -Anonymous reviewer 4 - Minor comment**

*L297: Phrase "P2 is be removed" is awkward. Please revise.*

This mistake is no longer found in the revised version of the manuscript.

**R4Mi12 -Anonymous reviewer 4 - Minor comment**

*L317: You state the instability of the CMI may be due to the interdependence of the ERT data. Could this be considered a strength? Since this means it can detect that these data were already inter-related and therefore do not function well as independent nodes in the network?*

See our previous answer: "We are not sure to understand how this can be a strength. We prefer to keep this as an explanatory hypothesis that could deserve further consideration. … "

Our hypothesis is better formulated in section 5.1.4, second paragraph (L420-425).

**R4Mi13 -Anonymous reviewer 4 - Minor comment**

*L329: Change "compare" to "compared"*

The discussion section has been entirely revised, and this mistake is no longer present.

**R4Mi14 -Anonymous reviewer 4 - Minor comment**

*Supporting information: There are some minor spelling mistakes in the document.*

The supplementary materials have been rechecked for grammar and spelling.

**R4Mi15 -Anonymous reviewer 4 - Minor comment**

*Figure SM2 caption: Are the descriptions for the symbols switched?*

Figure SM2 seems correct to us. The usual source of confusion is that CCM forecast direction is the opposite of the causal direction being tested. Another possible source of confusion is that, in the literature, causal dependencies are often found in the left (<0), because delays are reported according to a forecast horizon t+d. In our case, the causal dependencies are found on the right (positive d), since we test t-d.

---

## Author Response (AR2)

**Point by point reply to Anonymous Referee #1, Report #2**
* * *
1) Lines 348-350: my understanding is that the use of surrogates (e.g., bootstrapping) is only applied when we lack an analytical test of the null hypothesis. So, for the case of cross correlation, we don't need bootstrapping because there is an analytical null hypothesis. Please check.
* * *
We see two cases where the use of surrogate (or bootstrapping) techniques for significance testing are justified: (1) when we lack an analytical test (as mentioned); (2) when the underlying hypotheses the analytical test (e.g., normality and i.i.d) are not met (see Ebisuzaki, 1997). In the manuscript, we propose to use surrogate data testing to test the significance of correlation coefficients for the second reason, as hydrological variables are often skewed in their statistical distributions and serially correlated. Furthermore, if some analytical tests exist for the zero correlation, they may be misleading when applied to test for zero cross-correlation given the number of trials (lags involved in the computation of the correlogram). For all these cases, our point is that the surrogate data test is more generically applicable, at the cost of computational power, and less prone to false positives due to chance because it is permissive for violations of the usual normality and iid assumptions.

In addition, note that surrogate data tests are also used for the CCM method to test the significance of the Pearson's correlation coefficient measuring the predictive skills (e.g., van Nes et al. 2015).

We have modified the manuscript to include Ebisuzaki (1997), whose abstract reflects our point.

References:

Ebisuzaki, W. (1997). A Method to Estimate the Statistical Significance of a Correlation When the Data Are Serially Correlated, Journal of Climate, 10(9), 2147-2153. https://doi.org/10.1175/1520-0442(1997)010<2147:AMTETS>2.0.CO;2

van Nes, E., Scheffer, M., Brovkin, V. et al (2015). Causal feedbacks in climate change. Nature Clim Change 5, 445–448. https://doi.org/10.1038/nclimate2568
* * *
2) Lines 7, 62, 232, 299 and elsewhere: replace "study case" with "case study".
* * *
Modified as suggested
* * *
3) Line 270: replace "Result" with "Results"
* * *
Modified as suggested